# Evaluating Forecast Skills of Moisture from Convective-Permitting WRF-ARW Model during 2017 North American Monsoon Season

**Christoforus Bayu Risanto [1],\*, Christopher L. Castro [1], James M. Moker Jr. [2], Avelino F. Arellano Jr. [1], David K. Adams [3] , Lourdes M. Fierro [1] and Carlos M. Minjarez Sosa [4]**

[1] Department of Hydrology and Atmospheric Sciences, University of Arizona, Tucson, AZ 85721, USA; clcastro@email.arizona.edu (C.L.C.); afarellano@email.arizona.edu (A.F.A.J.); lourdesfierro@email.arizona.edu (L.M.F.)

[2] 56th Operations Support Squadron Luke Air Force Base, Glendale, AZ 85309-1215, USA; james.moker@us.af.mil

[3] Centro de Ciencias de la Atmósfera, Universidad Nacional Autónoma de México, 04510 Mexico City, Mexico; dave.k.adams@gmail.com

[4] Departamento de Fisica de la Universidad de Sonora, Hermosillo, Sonora 83000, Mexico; carlos.minjarez.s@gmail.com

\* Correspondence: cbrisanto@email.arizona.edu; Tel.: +1-520-621-0275

**Abstract:** This paper examines the ability of the Weather Research and Forecasting model forecast to simulate moisture and precipitation during the North American Monsoon GPS Hydrometeorological Network field campaign that took place in 2017. A convective-permitting model configuration performs daily weather forecast simulations for northwestern Mexico and southwestern United States. Model precipitable water vapor (*PWV*) exhibits wet biases greater than 0.5 mm at the initial forecast hour, and its diurnal cycle is out of phase with time, compared to observations. As a result, the model initiates and terminates precipitation earlier than the satellite and rain gauge measurements, underestimates the westward propagation of the convective systems, and exhibits relatively low forecast skills on the days where strong synoptic-scale forcing features are absent. Sensitivity analysis shows that model *PWV* in the domain is sensitive to changes in initial *PWV* at coastal sites, whereas the model precipitation and moisture flux convergence (QCONV) are sensitive to changes in initial *PWV* at the mountainous sites. Improving the initial physical states, such as *PWV*, potentially increases the forecast skills.

**Keywords:** weather research and forecasting model; convective-permitting parameterizations; global forecast system model; North American Mesoscale model; North American Monsoon precipitation; precipitable water vapor; moisture flux convergence; Global Positioning System; forecast skills of moisture; sensitivity analysis

## 1. Introduction

The North American monsoon season (NAMS) in northwestern Mexico and the southwestern United States generates severe thunderstorms with extreme precipitation. There are several unique natural hazards associated with the monsoon thunderstorms. Flash flooding and landslides occur due to the low moisture capacity of the soil and the generally steep structure of the terrain in the region [1,2]. Microbursts with high winds near the surface exceeding 25 m s$^{-1}$ may damage the natural landscape and built infrastructure [3]. Monsoon precipitation generally occurs from late June to mid-September, accounting for approximately 70% of the total annual rainfall in the western foothills of Sierra Madre

Occidental (SMO), including the Mexican states of Sonora, Sinaloa, and Nayarit [4], and 30% to 50% of annual rainfall in Arizona and New Mexico [5]. North American monsoon precipitation is a critical source of water for human and natural systems in this region [1,2,6].

Past work [7,8] has aimed to improve the North American monsoon precipitation forecast using convective-permitting modeling for retrospective numerical weather prediction (CPM NWP). Prior studies concluded that a CPM NWP could reasonably represent extreme monsoon precipitation events, especially in situations where mesoscale convective organization is facilitated by synoptic-scale features. A numerical weather prediction model is likely to produce a better forecast of severe thunderstorms and extreme precipitation during NAMS if it can represent both the mesoscale and the synoptic-scale features favorable for convective development.

NAMS thunderstorm development on the mesoscale is closely related to the diurnal cycle of convection over complex topography [9,10]. The diurnal cycle of convection requires atmospheric moisture and thermodynamic instability. There is disagreement within the literature as to the exact sources of moisture for the NAMS. Adams and Comrie [4] point out that the Gulf of Mexico (GoM) contributes to the upper-level (700 hPa to 200 hPa) moisture, and the Gulf of California (GoC) to the lower-level (below 700 hPa) moisture. The latter, also known as the gulf surge, is driven by the diurnal surface heat low forming over southwest Arizona and southern California. It is an important ingredient for the development of convection in the southwest United States [11–13]. The gulf surge significantly impacts the *PWV* along the coastlines of the GoC [7], as it propagates northwestward along the GoC [14]. In this study, *PWV* is defined as the total water vapor in a given vertical column of the atmosphere per unit cross-sectional area. It is commonly expressed in the height of the water produced by the condensation of the total water vapor in the vertical column. However, Moker et al. [8] suggests the impacts of the gulf surge on the convective development over northwest Mexico is negligible. Other studies [6,15,16] have given novel insight on the recycling of soil moisture and evapotranspiration, with respect to NAMS precipitation. Hu and Dominguez [16], for example, found that terrestrial evaporation contributes around 40% of the moisture during NAMS.

Thermodynamic instability occurs in conjunction with daytime heating. Upslope surface flow in the late morning leads to the convective development at around noon, over the crest of SMO [17]. The convection propagates westward in the late afternoon [10,17]. The convection may grow and organize into mesoscale convective systems (MCSs), which would reach the GoC by late evening [10,18–20].

NAMS onset is generally marked by the changing of the upper-level (300–250 hPa) flow from westerly to easterly over northwest Mexico and the southwest United States, as the upper-level anticyclone, or monsoon high, shifts northward [4,5]. A key synoptic feature is an inverted trough (hereafter referred to as IV), which propagates on the southern flank of the monsoon high [21,22]. This mid- to upper-level tropospheric low is associated with the mid-latitude breaking of Rossby waves or tropical upper-troposphere troughs. An IV induces a mid-level cyclonic circulation that dynamically facilitates convective organization [19,23,24]. Monsoon precipitation is significantly enhanced, especially in areas subject to less frequent monsoon precipitation driven by MCSs, such as west of the crest of SMO and the Mogollon Rim. The total precipitation doubles during the days with IVs, compared to the days without IVs [22,24].

Previous studies [25,26] have suggested applying high-resolution models for North American monsoon precipitation forecasts. The work of Moker et al. [8] shows the importance of resolving atmospheric features within the meso-$\gamma$ scale (1 to 4 km) using convective-permitting modeling. At this scale, monsoon convective structures, such as squall lines and outflow boundaries, can be realistically represented, similar to other parts of the world [27]. Even though the size of MCSs could be in the order of 1000 km, development depends on the meso-$\gamma$ scale features and requires less than 4 km grid spacing to reasonably represent them.

The North American Monsoon Global Navigational Satellite System (GNSS)/Global Positioning System (GPS) Transect Experiment 2013 (hereafter referred to as Transect 2013) collected *PWV* data

derived from nine GPS receiver antennas over northern Mexico. The Transect 2013 was in part motivated by the work of Kursinski et al. [28], which shows that convective precipitation NWP forecasts over northwestern Mexico are sensitive to the initial specification of *PWV*. The work of Moker et al. [8] found that retrospective NWP simulations using the Advanced Research version of the Weather Research Forecasting (WRF-ARW) model exhibit a moist bias in the initial *PWV* compared to observed *PWV*, and that the days with the presence of transient, upper-tropospheric IVs have higher modeled precipitation forecast skill than the days without the presence of IVs, especially over the slopes of northern SMO.

A second field campaign, the North American Monsoon GPS Hydrometeorological Network (hereafter referred to as GPS Hydromet 2017), was conducted from late June to mid-September 2017. The field campaign collected monsoon-related meteorological data, including GPS-derived *PWV*, over northwest Mexico and southwest United States. Based on the data of GPS Hydromet 2017, we set two objectives for this study: first, to evaluate the forecast skills of moisture from the convective-permitting WRF model in simulating the monsoon precipitation events of 2017, similar to the analyses by Moker et al. [8] in Transect 2013; second, to examine the sensitivity of the model-equivalent *PWV*, precipitation, and QCONV across the domain, especially with respect to the presence and the absence of IV, as well as to the different elevations of the sites. Other types of simulated hydrometeors were not included in our study, due to lack of observations to verify them.

## 2. Data and Model Description

### 2.1. Overview of the Data Collection and Ground Reference

As mentioned in the previous section, the meteorological data used in this study were collected during the GPS Hydromet 2017 field campaign. There are 20 observation sites installed in the area; 12 temporary field observation sites during the campaign period, indicated by circles in Figure 1, and eight permanent ones that belong to the Trans-boundary, Land and Atmosphere Long-term Observational and Collaborative Network (TLALOCNet), marked by squares in the same figure. The network of these observation sites covers elevations from sea level to over 2000 m above mean sea level (msl) in four different Mexican states: Sonora, Sinaloa, Chihuahua, and Baja California. Surface meteorological data gathered from these sites include atmospheric pressure, surface temperature, relative humidity, precipitation, wind speed, and wind direction with one-minute temporal resolution. Four stations, OPDE, RAYN, REF1, and TSFX, failed to record precipitation data during the field campaign.

Eighteen observation sites were equipped with GPS meteorological sensors/receivers, from which the zenith total delay (ZTD) estimate can be retrieved. With dual-frequency receivers, the delay of the signal traveling from the satellite to the receiving antenna is related to the total moisture content in the troposphere [29,30]. *PWV* can be estimated at 5-min intervals using the GNNS-inferred positioning system (GIPSY) processing technique, developed by the Jet Propulsion Laboratory (JPL). The integrated *PWV* is a function of the ZTD estimate, along with surface pressure and temperature measurements [31–33]. The advantages of GPS *PWV* are high temporal resolution and the ability to continuously observe in all weather conditions. In particular, GPS Transect 2013 and dense networks are able to temporally resolve monsoon convective development and organization, and so may be useful for data assimilation in CPM NWP [7,33–35]. Details of the retrieval technique and the GIPSY processing data are found in Moore et al. [30] and Fernandes et al. [32], respectively.

For purpose of model evaluation, we add seven precipitation datasets with 10-min temporal resolution from Comisión Nacional del Agua (CONAGUA) meteorological stations, indicated by triangles in Figure 1, as part of the field campaign. CONAGUA observation sites are located in Sonora. With these additional precipitation data, the study utilizes 23 observation sites equipped with rain gauge measurements. Details of observation site information are listed in Table 1.

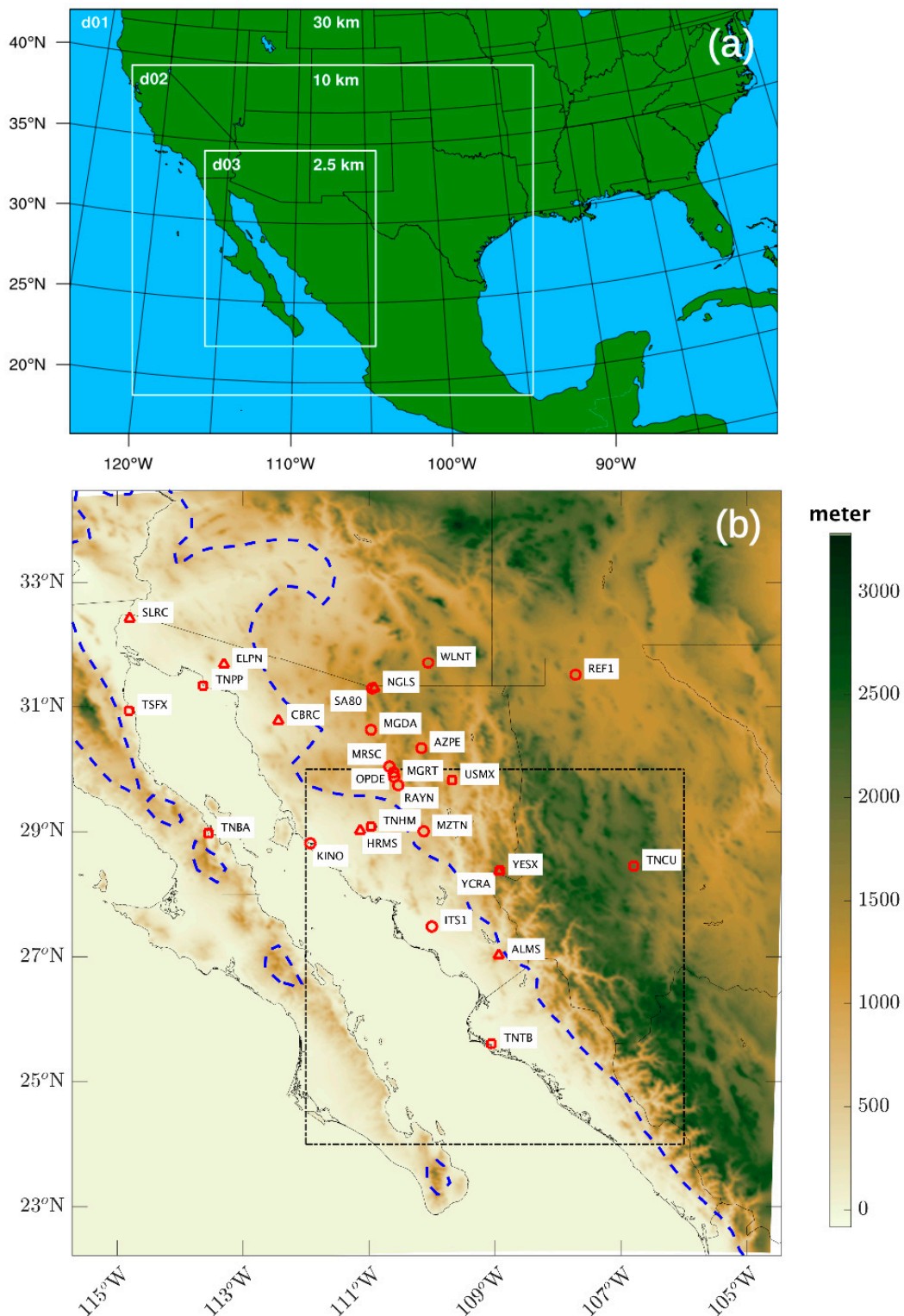

**Figure 1.** The Advanced Research version of the Weather Research Forecasting (WRF-ARW) nested domain configuration (**a**). The inner domain with plotted site locations (**b**). There are 12 temporary observation sites (circles), 8 observation sites of Trans-boundary, Land and Atmosphere Long-term Observational and Collaborative Network (TLALOCNet) (squares), and 7 observation sites of Comisión Nacional del Agua (CONAGUA) (triangles). Terrain elevation is shaded from 0 m to about 3000 m above mean sea level. The monsoon core region (24°–30° N and 112°–106° W) set up by NAME 2004 is outlined in black dash lines. Elevation cutoff at 500 m is outlined in a blue dashed line, separating the mountainous sites (>500 m) from the coastal sites (<=500 m).

**Table 1.** List of observation sites during GPS Hydromet 2017 field campaign. The list includes station names, latitudes, longitudes, elevations, managing institutions, and observed data available. The managing institutions include Comisión Nacional del Agua (CONAGUA) and Trans-boundary Land and Atmosphere Long-term Observational and Collaborative Network (TLALOCNet).

| Station | Lat (N) | Lon (W) | Elev (m msl) | Institution | PWV | Precipitation |
|---------|---------|---------|--------------|-------------|-----|---------------|
| ALMS | 27.0217 | 108.9378 | 407.00 | CONAGUA | no | yes |
| AZPE | 30.3369 | 110.1663 | 838.00 | temporary | yes | yes |
| CBRC | 30.7719 | 112.4353 | 201.00 | CONAGUA | no | yes |
| ELPN | 31.6800 | 113.3047 | 131.00 | CONAGUA | no | yes |
| HRMS | 29.0133 | 111.1369 | 150.00 | CONAGUA | no | yes |
| ITS1 | 27.4845 | 110.0000 | 31.00 | temporary | yes | yes |
| KINO | 28.8149 | 111.9287 | 0.00 | temporary | yes | yes |
| MGDA | 30.6321 | 110.9676 | 755.00 | temporary | yes | yes |
| MGRT | 29.8762 | 110.5964 | 625.00 | temporary | no | yes |
| MRSC | 30.0404 | 110.6737 | 721.00 | temporary | no | yes |
| MZTN | 29.0030 | 110.1300 | 544.00 | temporary | yes | yes |
| NGLS | 31.2978 | 110.9139 | 1291.00 | CONAGUA | no | yes |
| OPDE | 29.9444 | 110.6121 | 690.00 | temporary | yes | no |
| RAYN | 29.7410 | 110.5366 | 635.00 | temporary | yes | no |
| REF1 | 31.5112 | 107.7173 | 1227.00 | temporary | yes | no |
| SA80 | 31.2934 | 110.9465 | 1274.00 | temporary | yes | yes |
| SLRC | 32.4239 | 114.7978 | 37.00 | CONAGUA | no | yes |
| TNBA | 28.9719 | 113.5473 | 5.00 | TLALOCNet | yes | yes |
| TNCU | 28.4506 | 106.7940 | 2111.00 | TLALOCNet | yes | yes |
| TNHM | 29.0813 | 110.9703 | 202.00 | TLALOCNet | yes | yes |
| TNPP | 31.3355 | 113.6316 | 39.00 | TLALOCNet | yes | yes |
| TNTB | 25.6059 | 109.0527 | 78.00 | TLALOCNet | yes | yes |
| TSFX | 30.9339 | 114.8106 | 28.00 | TLALOCNet | yes | no |
| USMX | 29.8217 | 109.6810 | 656.00 | TLALOCNet | yes | yes |
| WLNT | 31.7057 | 110.0575 | 1411.00 | temporary | yes | yes |
| YCRA | 28.3667 | 108.9333 | 1551.00 | CONAGUA | no | yes |
| YESX | 28.3783 | 108.9196 | 1537.00 | TLALOCNet | yes | yes |

Four satellite-derived precipitation products provide gridded precipitation estimates: (i) Integrated Multi-satellitE Retrievals for Global Precipitation Measurement Early Precipitation L3 Half Hourly 0.1 degree x 0.1 degree V05 (GPM_3IMERGHHE V05; referred to as GPM Early), and (ii) GPM_3IMERGHH V05 (referred to as GPM Final), with similar spatial and temporal resolutions [36,37]; (iii) the National Oceanic and Atmospheric Administration (NOAA) Climate Prediction Center morphing technique (CMORPH) [38] with 8 km/30 min resolution; (iv) and the Precipitation Estimation from Remotely Sensed Information using Artificial Neural Networks (PERSIANN) [39] with $0.25 \times 0.25$ degree and 1 h resolution. Details on all satellite-derived precipitation products are provided in Table 2. The field campaign data and the satellite data are available on http://hidromet-data.unison.mx/u/login upon request.

**Table 2.** List of satellite-based precipitation products used in the study.

| Product | Source | Spatial Resolution | Temporal Resolution | References |
|---------|--------|--------------------|--------------------|------------|
| GPM Early | NASA | 0.1° | Half-hourly | Huffman (2017) [36] Hou et al. (2014) [37] |
| GPM Final | NASA | 0.1° | Half-hourly | Huffman (2017) [36] Hou et al. (2014) [37] |
| CMORPH | NOAA | 8 km | Half-hourly | Joyce et al. (2004) [38] |
| PERSIANN | UCI | 0.25° | 1 hourly | Sorooshian et al. (2002) [39] |

There are notable differences among these satellite-derived precipitation products in comparison to rain gauge data. We compute the root mean square error (RMSE) and biases between the daily rain gauge and the satellite-based precipitation measurements from each observation site from 30 June to 12 September. Since the observation site is not always at the center of the grid point in the satellite products, we use inverse distance weighting technique to estimate the precipitation value. The result is shown in Figure 2.

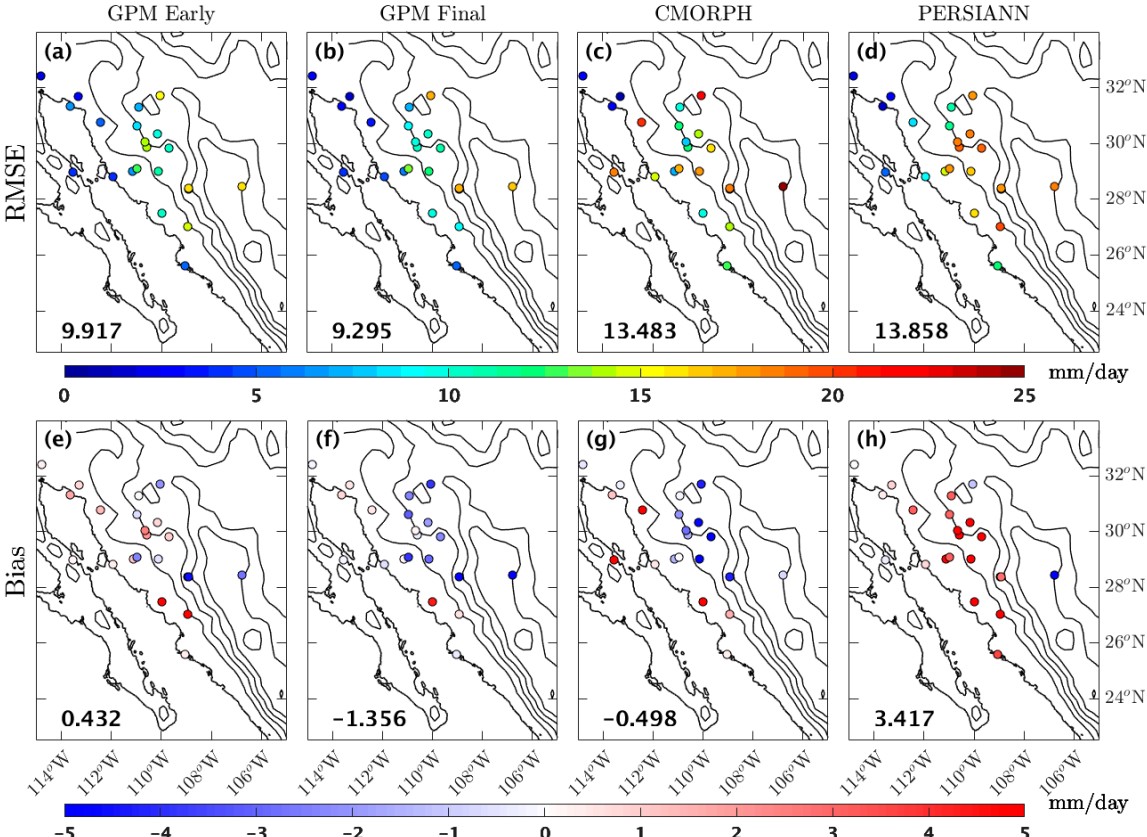

**Figure 2.** Root mean square error (RMSE) and bias between the daily rain gauge measurements at the observation sites and the satellite-based daily precipitation measurement from 30 June to 12 September. Bias is calculated as satellite measurement minus rain gauge measurement. The value at the bottom left of panel (**a–d**) is the mean RMSE, and the value at the bottom left of panel (**e–h**) is the mean bias. Global precipitation measurement (GPM) Final has the lowest RMSE value; GPM Early has the lowest bias. Terrain is contoured every 500 m (black lines).

GPM Final has the lowest daily precipitation RMSE among the four products, whereas GPM Early has the lowest daily precipitation bias (Figure 2). The high elevation sites (>500 m) tend to have high RMSE of daily precipitation, relative to low elevation. Both GPM Final and CMORPH underestimate daily precipitation over the mountainous areas, as seen in their negative bias in Figure 2. The error in estimating precipitation over mountainous areas has been previously documented. A study in East Africa by Dinku et al. [40] shows CMORPH and Tropical Rainfall Measurement Mission (TRMM)-3B42 underestimate rainfall over the highlands of Ethiopia. The work of Mei et al. [41], finds that TRMM-3B42, CMORPH, and PERSIANN overestimate low precipitation and underestimate high precipitation accumulations over the eastern Italian Alps.

Our finding is similar to that of Zhang et al. [42], which showed that GPM Final outperformed CMORPH in estimating daily precipitation over regional and sub-regional scales and low and mid-elevations in the Tianshan mountains in China. Even though we did not include any TRMM product, many studies [42–44] have found that GPM Final performs better than TRMM, especially

with respect to daily precipitation and the spatial distribution of precipitation. Since TRMM daily precipitation estimates are also found to be better than that of PERSIANN (e.g., Zhou et al. [45]), we conclude that GPM Final was the best satellite-derived precipitation product available during the GPS Hydromet 2017 field campaign. Therefore, we used only the GPM Final precipitation product for evaluating the model simulations.

In addition, we used Geostationary Operational Environmental Satellite 15 Water Vapor (hereafter referred to as GOES-15 WV) imagery to understand the synoptic patterns of the upper-level atmospheric flow. This data are obtained from the NASA Langley Cloud and Radiation Research Group (http://www-angler.larc.nasa.gov). Water vapor data from CONUS GOES 4 km and surface analysis with GOES West IR imagery are additionally considered. An analysis product from Global Forecast System (GFS) Analysis is utilized to ascertain the wind patterns at 300 hPa, and potential vorticity (PV) anomalies at the dynamic tropopause, for detection of IVs. The WV imagery, the 300 hPa winds, and the PV anomaly, are examined prior to the period of monsoon precipitation to identify the characteristics of the monsoon day, explained in Section 3.1. Table 3 shows the details of these additional datasets and their purpose within the scope of this study.

**Table 3.** Other satellite and model products used for classifying monsoon days.

| Dataset | Sources | Purpose |
|---|---|---|
| Surface Meteorological data | http://mesowest.utah.edu | To identify the rainfall |
| GOES-15 Water Vapor | NASA Langley Cloud and Radiation Research Group (http://www-angler.larc.nasa.gov) | To identify upper-level atmospheric dynamics and inverted troughs (IVs) with convective development |
| GFS Analysis (0.5°/6-hourly, 31 vertical levels) | NCDC https://www.ncdc.noaa.gov/ | To identify the wind patterns at 300 hPa and the PV anomaly at 2-PVU layer |
| CONUS GOES 4-km water vapor imagery | https://mesonet.agron.iastate.edu/ GIS/goes.phtml | To identify IVs with convective development |
| Surface analysis and GOES-West IR imagery | https://www.wpc.ncep.noaa.gov/ #page=ovw | To identify early convective organization |

## 2.2. Convective-Permitting Model Configuration

The Advanced Research of the Weather Research and Forecasting (WRF-ARW) [46] model, version 3.9, is used for retrospective daily convective simulations (NWP hindcasts) for the duration of GPS Hydromet 2017. The configuration is based on the real time quasi-operational model in the University of Arizona, Department of Hydrology and Atmospheric Sciences (http://www.atmo.arizona.edu/ ?section=weather&id=wrf). The design of the experiment is very similar to that of Moker et al. [8], consisting of three nested domains (d01, d02, d03), as described in Figure 1 and Table 4.

**Table 4.** Horizontal resolution of the nested domains in our WRF-ARW experiments with different initial conditions and lateral boundary conditions. In the 2.5 km convective-permitting domain, we turn off cumulus parameterization. In the larger domains, we use Kain-Fritsch cumulus parameterization.

| WRF-ARW Experiments | d01 | d02 | d03 |
|---|---|---|---|
| WRF-GFS | 30 km | 10 km | 2.5 km * |
| WRF-NAM | 10 km | 2.5 km * | |

\* no Kain-Fritsch cumulus parameterization.

The vertical dimension has 27 levels with sigma coordinate, in which the hydrostatic pressure follows the topography of the terrain [46]. The inner domain is similar to that of Transect 2013, which is based on the North American Monsoon Experiment (NAME) 2004 Tier I region [17,25]. With 2.5 km

grid spacing, the inner domain is CPM resolution with convective parameterization scheme deactivated and the ability to resolve convective development in meso-$\gamma$ scale.

The 24-h simulation is initialized at 1200 UTC for 48 selected days during the GPS Hydromet 2017 period, with two different sets of lateral boundary forcing and initial conditions: (1) the Global Forecast System model (hereafter referred to as WRF-GFS) and (2) the North American Mesoscale model (hereafter referred to as WRF-NAM). The GFS model has a spatial resolution of 0.5°, and the NAM model has a spatial resolution of 12 km. Because of its spatial resolution, WRF-NAM uses only two domains, with 10 km and 2.5 km spatial resolutions. The complete list of physics schemes used in our models is provided in Table 5.

**Table 5.** List of the physics schemes used in the WRF-ARW configuration and applied to all domains.

| Category | Scheme | Reference |
|---|---|---|
| Microphysics | WRF single-moment 6-class | Hong and Lim (2006) [47] |
| Longwave radiation | Rapid Radiative Transfer Model | Iacono et al. (2008) [48] |
| Shortwave radiation | Goddard | Chou and Suarez (1999) [49] |
| Land Surface | Noah-MP (multi physics) | Niu et al. (2011) [50] |
| Planetary boundary layer | Yonsei University | Hong et al. (2006) [51] |

## 3. Analysis Methods

### 3.1. Classifying IV and Non-IV Days

The work of Moker et al. [8] classified monsoon precipitation based on synoptic and mesoscale features, and concluded that an IV is the most important feature for facilitating convective development. Applying the same criteria, we distinguish the days with monsoon precipitation from the days with monsoon break, light convection, or tropical cyclones (TC), and classify the monsoon precipitation days based on the presence and absence of IVs.

We use surface meteorological data, the WV imagery from GOES 15 and CONUS GOES 4-km, and GOES IR imagery to qualitatively identify the days with monsoon precipitation, within a region bounded by 26.0°–32.5° N and 114°–107.25° W. A monsoon 'precipitation day' is defined as when organized convection develops in the eastern slopes of SMO at around noon time LT (~1800 UTC), propagates off the high terrain of SMO and westward toward GoC, reaches a mature stage around late afternoon LT (~0000 UTC), and dissipates at late evening LT (~0600 UTC), and when the 24-h total precipitation within the region exceeds 20 mm. Days with precipitation in association with a tropical cyclone are excluded.

Based on Bieda et al. [24] and Moker et al. [8], we identify from the monsoon precipitation days subsets of the days with IVs and without IVs. We consider only the presence of IVs inside or approaching the monsoon core region (24–30° N and 106–112° W) [17,19,25]. GOES 15 WV and CONUS GOES WV imageries help detect the cyclonic pattern in the southern flank of the monsoon high. The GFS Analysis 300 hPa winds and pressure anomaly at 2-potential vorticity unit (PVU) confirm the presence of IVs. A day with presence of an IV is referred to as strongly forced day, and the one without is a weakly forced day. During the GPS Hydromet 2017 field campaign, we found nine strongly forced days and 39 weakly forced days. The other days are classified monsoon break, light convective, or TC Lidia. Table 6 summarizes the meteorological classification of all days during the field campaign.

**Table 6.** List of days during GPS Hydromet 2017 field campaign. Based on synoptic and mesoscale features, the days are classified into strongly and weakly forced days, as well as days with light convection, tropical cyclone, and monsoon break. There are nine strongly forced days, 39 weakly forced days, nine days with light convections, 16 days of monsoon break, and 2 days of precipitation associated with the TC Lidia.

| Weakly Forced Days | Strongly Forced Days | Light Convection | Break | TC |
|---|---|---|---|---|
| 1 Jul to 5 Jul | 7 Jul to 9 Jul | 30 Jun | 17 Jul | 1 Sep |
| 10 Jul to 12 Jul | 13 Jul | 6 Jul | 21 Jul | 2 Sep |
| 15 Jul | 14 Jul | 15 Aug | 30 Jul | |
| 16 Jul | 18 Jul | 20 Aug | 5 Aug | |
| 19 Jul | 27 Jul | 21 Aug | 6 Aug | |
| 20 Jul | 28 Jul | 24 Aug | 13 Aug | |
| 22 Jul to 26 Jul | 18 Aug | 10 Sep to 12 Sep | 14 Aug | |
| 29 Jul | | | 17 Aug | |
| 31 Jul | | | 28 Aug to 31 Aug | |
| 1 Aug to 4 Aug | | | 5 Sep | |
| 7 Aug to 12 Aug | | | 6 Sep | |
| 16 Aug | | | 8 Sep | |
| 19 Aug | | | 9 Sep | |
| 22 Aug | | | | |
| 23 Aug | | | | |
| 25 Aug to 27 Aug | | | | |
| 3 Sep | | | | |
| 4 Sep | | | | |
| 7 Sep | | | | |

We evaluate the model *PWV* by comparing the GPS-derived *PWV* with the model-equivalent GPS *PWV* in the sub-diurnal interval for weakly and strongly forced days. An inverse-distance squared weighting technique is used to extract the variables necessary to compute model-equivalent GPS *PWV* from the 18 observation sites in the WRF simulation output. Model *PWV* is computed as:

$$PWV_{i,j} = \frac{1}{\rho_w \, g} \sum_{k=1}^{nk-1} \frac{1}{2} (QV_k + QV_{k+1})(p_k - p_{k+1}) \tag{1}$$

Here $p$ is pressure (Pa), including surface pressure ($PSFC$), assigned as $p_1$; $\rho_w$ is water density ($1000 \, \text{kg m}^{-3}$); $g$ is gravity ($9.81 \, \text{m s}^{-2}$); and $QV$ is specific humidity (kg water vapor/kg air), including specific humidity at 2 m ($Q2$), assigned as $QV_1$. This variable is defined as $\frac{QVAPOR}{1+QVAPOR}$, where $QVAPOR$ comes from the model output. The $k$ and $nk$ represent the model level and the number of model levels (i.e., 28) respectively. The unit of *PWV* is mm. The diurnal cycle of model *PWV* and GPS-derived *PWV* across observation sites for the weakly and strongly forced days, as well as the combined weakly–strongly days are then computed along, with the standard error of the observation. These model *PWV* and GPS-derived *PWV* are compared in terms of magnitude and phase. To make an appropriate comparison between the *PWV* and precipitation diurnal cycles, the computation includes only 14 sites, which have both *PWV* and precipitation data.

*3.2. Forecast Skill Evaluation*

To evaluate the model precipitation, we compute the biases between the model precipitation, which is the non-convective rain (RAINNC) variable from the convective-permitting domain of the WRF, and the precipitation calibrated of the GPM Final product. Since the computation counts on every grid point within the domain, we scale up the model-equivalent precipitation grids from 2.5 km horizontal resolution to 0.1° using the Earth System Modeling Framework (ESMF) "conserve" function available in National Center for Atmospheric Research Command Language (NCL). In order to see the

sub-diurnal development, we calculate the 3- and 24-hourly total precipitation and normalize it into a mean hourly precipitation rate.

To measure the accuracy of the model-equivalent precipitation, categorical statistics are used. These verifications include Critical Success Index (*CSI*) [52], Probability of Detection (*POD*), and False Alarm Ratio (*FAR*), as described in Wilks [53]. The mathematical definition of each is as follows:

$$CSI = \frac{Hits}{Hits + False\ Alarm + Misses} \tag{2}$$

$$POD = \frac{Hits}{Hits + Misses} \tag{3}$$

$$FAR = \frac{False\ Alarm}{Hits + False\ Alarm} \tag{4}$$

A precipitation event is defined as the total accumulation of at least 2.5 mm (10 mm) in a grid point in the 6-hourly (daily) periods. The *CSI*, *POD*, and *FAR* values range from 0 to 1. As *CSI* and *POD* approach 1 (0), the WRF precipitation forecast skill increases (decreases). On the other hand, as *FAR* approaches 0 (1), the precipitation forecast skill increases (decreases).

In order to assess the WRF precipitation forecast skill between the strongly and the weakly forced days, we subtract the *CSI*, *POD*, and *FAR* values of each grid in the weakly forced days from those in the strongly forced days, similar to Moker et al. [8]. The statistical field significance is computed with a Monte Carlo technique, as described by Livezey and Chen [54]. Using 1000 permutations, we resample the strongly and weakly forced days randomly and compute the statistical verifications. The critical value of the statistical field significance is set to 90%, which is the 900th value of the histogram. We examine both the 24-hour total precipitation and the 6-hour total precipitation.

### 3.3. Sensitivity Analysis

Finally, we conduct a sensitivity analysis of WRF-GFS *PWV*, precipitation, and QCONV of one weakly and one strongly forced day. The goal is to predict changes in the variance of these three metrics, given a set of ensemble forecasts [55,56]. For this purpose, we first apply the CV3 background error covariance option in the WRF Data Assimilation system [8,57,58] to add small perturbations on the outer most domain (d01) of the model meteorological fields at 0000 UTC, in order to create 20 ensemble members. We spin them up for 6 hours to obtain appropriate variance of state variables in d03. Appropriate variance is achieved when the growth of the ensemble spread in the core domain (d03) across time stabilizes. We find that it stabilizes in 6 to 12 hours. The variables being perturbed are stream function, unbalanced velocity potential, unbalanced temperature, specific humidity, and unbalanced surface pressure. Then, we integrate the model for another 6 hours to propagate the perturbations to the inner domain, so as to generate the initial condition. Each ensemble member is then run deterministically for 24 hours starting at 1200 UTC, to generate an hourly forecast.

The sensitivity analysis is computed using regression, as follows:

$$\left( \frac{\Delta X_{i,j,t+fh}}{\Delta PWV_{isite,t=0}} \right) = \frac{cov\left(X_{i,j,t+fh}, PWV_{isite,t=0}\right)}{var\left(PWV_{isite,t=0}\right)} \times \frac{ens\ mean\left(PWV_{isite,t=0}\right)}{ens\ mean\left(X_{i,j,t+fh}\right)} \tag{5}$$

where $X_{i,j,t+fh}$ represents a variable from the ensemble WRF-GFS at grid point (*i,j*), the subscript $t + fh$ indicates the forecast hour (*fh*), and $PWV_{isite,t=0}$ is the model-equivalent GPS *PWV* for each ensemble at each site at the initial condition. We interpret the results as the percent of change of a particular variable (or the local linear sensitivity) in the domain at the later forecast hours, to the change of initial *PWV* at a particular site [8]. By omitting the last terms, i.e., $\frac{ens\ mean\left(PWV_{isite,t=0}\right)}{ens\ mean\left(X_{i,j,t+fh}\right)}$, we have the results as the change in the variable at the later forecast hours to the change of initial *PWV* at a particular site, with the unit of the variable per mm of *PWV*.

We compute the sensitivity analysis for *PWV*, precipitation, and QCONV per hour for each site. In relation to *PWV* and precipitation, QCONV, or moisture flux convergence, is a measure of horizontal and vertical water vapor convergence, as well as its advection in the atmosphere [59]. These terms can be derived from the conservation of water vapor equation. Even though QCONV is a diagnostic measure [kg kg$^{-1}$ s$^{-1}$], it is often used to help predict convective initiation in operational forecasting [59]. In this study, the grids for the QCONV computation are scaled up from 2.5 km to 1/16° to smooth out the results.

We classify the sites into coastal and mountainous areas, and we take the mean of the sensitivity across the sites for each class. To quantitatively compare the sensitivity of the mountainous with that of the coastal areas, the mean of the absolute values across the domain is calculated for each hour. The reason for this geographical classification is that the character of convective precipitation is different. Mountainous topography assists convective organization due to orographic lifting [60,61]. Therefore, the frequency of precipitation in mountainous areas is generally higher than lowlands. The definition of coastal and mountainous sites is based on previous studies that use elevation-based threshold criteria. The work of Massmann et al. [62] states that a mountainous area in mid-latitudes is defined as a land with elevation ranging from 500 to 1500 m msl. Barry [63] specifies that a mountain region be above 600 m msl. The work of Luong et al. [64] shows there is an elevation transition point of around 1000 m in association with mountain–valley convective precipitation, generated by a modified Kain and Fritsch convective scheme. Ideally, if we had more sites, we would have classified the domain into three classes, i.e., mountainous, coastal, and low-lying land. Due to the small number of sites, we classify the domain into two classes, i.e., mountainous and coastal areas. We define a mountainous area as any land with elevation higher than 500 m msl., and a coastal area as any elevation less or equal to 500 m msl, as shown in Figure 1b. There are seven GPS sites within the coastal area, and 11 GPS sites located within the mountainous area.

In addition to the calculation of mean across the mountain and coastal sites, a covariance-based Empirical Orthogonal Function (EOF) analysis is performed. The samples are all the sensitivity analysis results of *PWV* for each site. We calculate the Eigen spectrum of the sensitivity to evaluate the variance explained. Significant differences in explained variance between the first two or three principal components indicate the existence of significant dominant modes or patterns. The regression maps are plotted by multiplying the first principal component with the data anomaly. The purpose is to confirm the dominant mode of the sensitivity at each forecast hour, and the sites with the strongest influence on the changes of *PWV* in the domain at the later forecast hours. We hypothesize that the dominant pattern in the sensitivity field should reflect the difference between coastal and mountain sites.

## 4. WRF Performance Evaluation

In our study, we simulate each day that is classified as a strongly or weakly forced day during the GPS Hydromet 2017 field campaign, per the criteria of Moker et al. [8]. The simulation is initialized at 1200 UTC (0500 LT) and runs for 24 hours. The following sub-sections are the evaluation of the model performance and skill.

### 4.1. PWV and Precipitation Diurnal Cycles

In general, the WRF-GFS overestimates *PWV* diurnal cycle, no matter whether the event is classified as a weakly or strongly forced day (Figure 3a–c). However, at the last four hours of the forecast period, WRF-GFS *PWV* values are drier than the GPS-derived *PWV* values. Both for the combined and weak cases, the WRF-GFS *PWV* values are around 0.6 mm less than the GPS-derived *PWV* values. Even though the difference between the model and the observations is obvious, most of the time the WRF-GFS *PWV* is within the standard error of the GPS-derived *PWV* measurement, as shown by the shaded areas, except in the period of 0100 to 0400 UTC (1800–2100 LT). This is the time when diurnal precipitation is at its peak, as shown in Figure 3d–f.

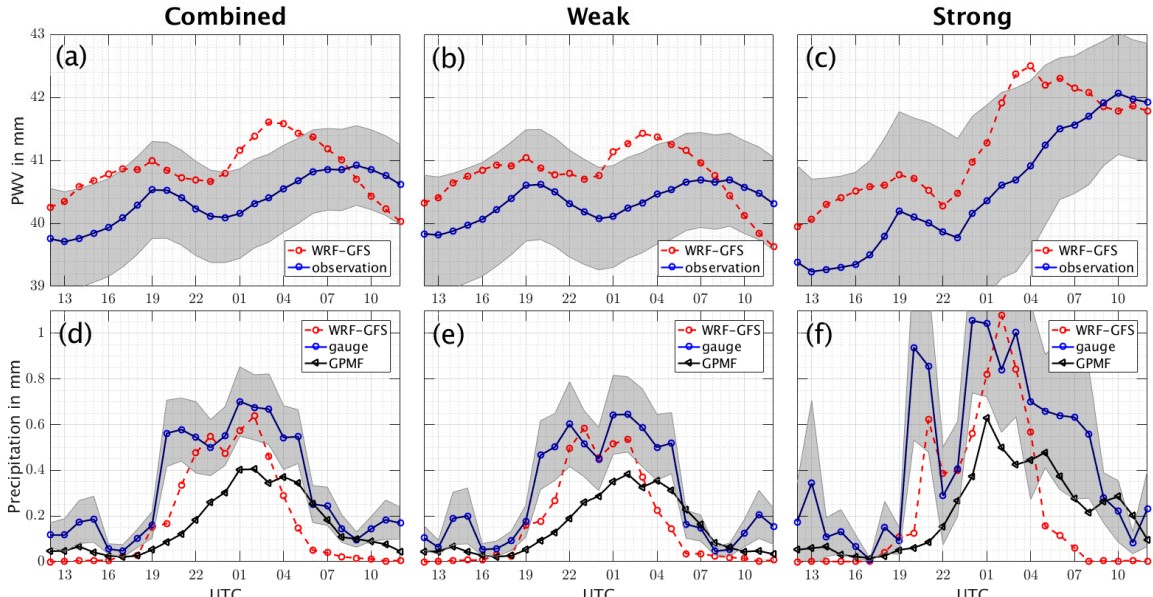

**Figure 3.** Mean diurnal cycle of WRF-GFS *PWV* and GPS-derived *PWV* (or observations) across 14 stations for combined strongly—weakly forced days (**a**), weakly forced days (**b**), and strongly forced days (**c**). Mean diurnal cycle of precipitation across 14 stations, based on WRF-GFS simulations, rain gauge measurement, and GPM Final product for combined strongly–weakly forced days (**d**), weakly forced days (**e**), and strongly forced days (**f**). Shaded areas in grey represent standard errors of the observations.

The two maxima in the *PWV* diurnal cycle correspond to the two peaks of precipitation. When precipitation occurs, the atmosphere is saturated with moisture from rainwater droplets and cloud droplets. The two maxima occur because of the timing difference in precipitation peak at some sites. The first *PWV* maximum is associated with the first peak of precipitation, occurring around 2000 to 2200 UTC (1300–1500 LT) at MGDA and SA80 (Figures S1 and S2). The second *PWV* maximum, which occurs between 0000 to 0400 UTC (1700 to 2100 LT), is associated with the second peak of precipitation, similar to the finding by Nesbitt et al. [10], which shows that a precipitation peak occurs between 1800 to 2000 LT.

There is a timing issue in the WRF-GFS simulations. We note in Figure 3a that the WRF-GFS *PWV* values decrease to the minimum, from 40.99 mm to 40.66 mm, within four hours of 1900 UTC (1200 LT) to 2300 UTC (1600 LT), while the GPS-derived *PWV* values decrease from 40.53 mm to its minimum, 40.09 mm, in five hours from 1900 UTC (1200 LT) to 0000 UTC (1700 LT). As a result, the WRF-GFS simulations reach the *PWV* minimum an hour earlier than the GPS-derived *PWV* does, creating an hour lag between the WRF-GFS *PWV* diurnal cycle and the GPS-derived *PWV* diurnal cycle. This holds true for weakly forced days (Figure 3b) and strongly forced days (Figure 3c).

In Figure 3a, the WRF-GFS simulations rapidly increase the *PWV* values from 40.66 mm at 2300 UTC (1600 LT) to maximum 41.60 mm at 0300 UTC (2000 LT) while the observations exhibit a slower increase over a longer time period, from 40.09 mm at 0000 UTC (1700 LT) to maximum 40.92 mm at 0900 UTC (0200 LT). As a result, the WRF-GFS *PWV* diurnal cycle reaches the second maximum around six hours earlier than the GPS-derived *PWV*. Thus, the WRF-GFS *PWV* becomes more out of phase, compared with the GPS-derived *PWV* diurnal cycle, during the late afternoon to evening hours. At this time, WRF-GFS *PWV* values go beyond the range of the GPS-derived *PWV* standard error. Similar time lags and overestimation are also observed both in the weakly forced days (Figure 3b) and the strongly forced days (Figure 3c).

The only noticeable difference between the weakly and strongly forced days is the amount of hourly WRF-GFS *PWV* in the last six hours of the forecast. The weakly forced days (Figure 3b) exhibit a rapid decrease, from 41.40 mm to 39.60 mm, in the last six hours of forecast. The final WRF-GFS *PWV* value is almost out of the range of standard error. The strongly forced days (Figure 3c), on the

other hand, show a slow decrease from 43.50 mm to 41.75 mm, within the same period. The final WRF-GFS *PWV* value is within the range of standard error and close to the final GPS-derived *PWV* value. The difference is greater on the weakly forced days than on strongly forced days.

The rapid decrease in WRF-GFS *PWV* corresponds to the early termination of convective precipitation (Figure 3d–f). The WRF-GFS precipitation for all cases is within the standard error of the rain gauge measurement, from about 2200 to 0200 UTC (1500 to 1900 LT). The modeled values are closer to the rain gauge than to the GPM Final product, whose values fall outside the range of standard error of the rain gauges. After 0200 UTC (1900 LT), the amount of precipitation decreases rapidly, from 0.65 mm at 0200 UTC (1900 LT) to less than 0.1 mm at 0600 UTC (2300 LT), and exceeds the standard error, just as the WRF-GFS *PWV* decreases rapidly (Figure 3a). This behavior is basically the same for the weakly and strongly forced days. While the rain gauge and the GPM Final still indicate ongoing precipitation after 0500 UTC (2200 LT), the amount of WRF-GFS precipitation is close to zero. This low value marks the end of the convective precipitation in the simulations. This early termination issue in the WRF-GFS precipitation is consistent with Moker et al. [8].

Unlike in the WRF-GFS simulations, the WRF-NAM simulations overestimate the *PWV* diurnal cycle throughout the forecast period, so that their values exceed the standard error (Figure 4a–c). This high bias holds true for the weakly and strongly forced days. At the initial forecast hour, the mean difference is more than 3.00 mm for weakly forced days, strongly forced days, and the combination of both. At the end of the forecast hour, the difference is around 1.00 mm for all cases. The WRF-NAM *PWV* diurnal cycle contains only one minimum and one maximum. The maximum in the first 12 h of forecast is not well defined.

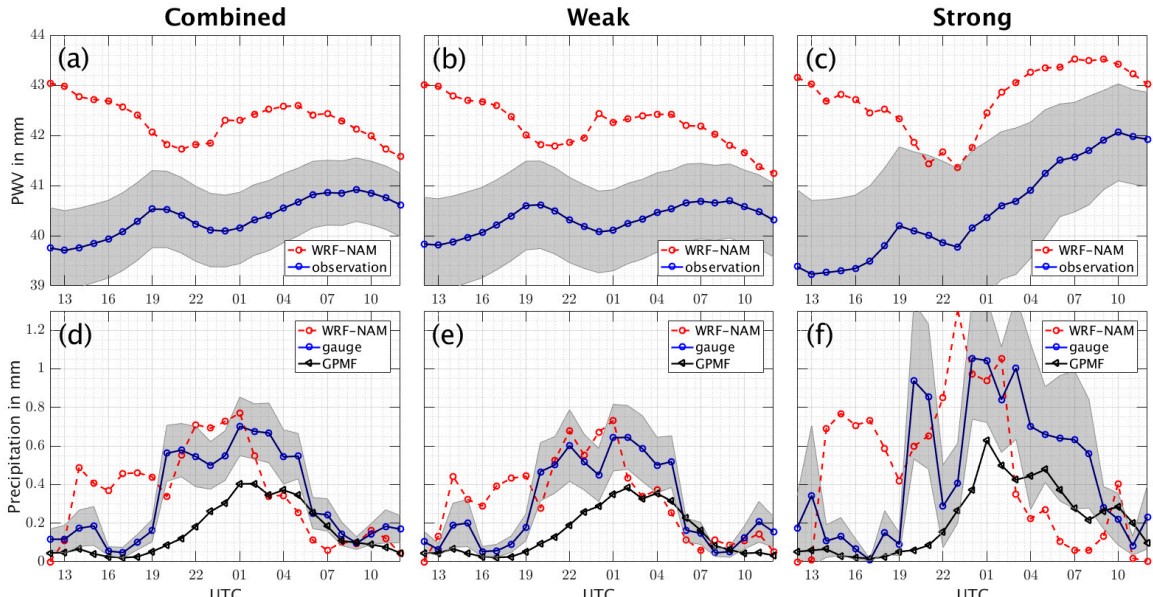

**Figure 4.** Mean diurnal cycle of WRF-NAM *PWV* and GPS-derived *PWV* (or observations) across 14 stations for combined strongly-weakly forced days (**a**), weakly forced days (**b**), and strongly forced days (**c**). Mean diurnal cycle of precipitation across 14 stations, based on WRF-NAM simulations, rain gauge measurement, and GPM Final product for combined strongly–weakly forced days (**d**), weakly forced days (**e**), and strongly forced days (**f**). Shaded areas in grey represent standard errors of the observations.

Similar to the WRF-GFS, the minimum and the maximum of the *PWV* diurnal cycle in the WRF-NAM are out of phase compared with those of the GPS-derived *PWV* diurnal cycle, due to rapid decrease and rapid increase within 12 h of forecast. The WRF-NAM *PWV* in Figure 4a reaches the minimum of 41.72 mm at 2100 UTC (1400 LT), compared with the GPS-derived *PWV* minimum at 0000 UTC (1700 LT), as shown earlier. Thus, it creates a three-hour lag within the 12-h forecast. The strongly forced days (Figure 4c) also exhibit higher *PWV* values throughout the diurnal cycle than the weakly

forced days or the combination of both. The difference between the initial WRF-NAM *PWV* value and the GPS-derived *PWV* value almost reaches 4.00 mm. However, the strongly forced days do not exhibit any time lags, since the minimum of WRF-NAM *PWV* and GPS derived *PWV* diurnal cycle occurs at 2300 UTC (1600 LT).

The high bias of WRF-NAM *PWV* generates precipitation as early as the first forecast hour at 1300 UTC (0600 LT), as shown in Figure 4d–f. This model precipitation exceeds the standard error of the rain gauge measurements. The WRF-NAM precipitation values come to the standard error range at 2100 to 0100 UTC (1400 to 1800 LT), even though overestimation occurs within the period. Similar to the WRF-GFS simulations, this model, in all cases, also exhibits rapid decrease in precipitation after 0200 UTC (1900 LT), marking early termination of convective precipitation.

In summary, both the WRF-GFS and WRF-NAM simulations generate higher values in the *PWV* diurnal cycle at the initial forecast hour than the GPS-derived *PWV*. This result agrees with that of Moker et al. [8], where wet bias was present in Transect 2013. While the WRF-GFS *PWV* is generally within the range of the standard error of observations, WRF-NAM *PWV* is out of range of the standard error. The time lags and the rapid decrease of WRF-GFS *PWV* result in early termination of the WRF-GFS precipitation. It is also notable that WRF-GFS precipitation is closer to rain gauge measurements than to satellite precipitation products. The overestimation of WRF-NAM *PWV* generates an early initiation and termination of modeled precipitation, with high positive bias.

*4.2. Precipitation Diurnal Cycle in the Domain*

While Figures 3d–f and 4d–f show the mean diurnal cycles of precipitation across 14 observation sites, Figures 5 and 6 exhibit the mean diurnal cycle of precipitation rate across the region (first row) and its bias (third row), with respect to the GPM Final precipitation products (second row). The display of these figures and the bias calculation are similar to those in Moker et al. [8]. The difference is that the study by Moker et al. [8] finds no statistically significant bias between the strongly and weakly forced days in Transect 2013. In our study, the bias at 0900–1200 UTC between the strongly and weakly forced days is statistically significant in the WRF-GFS simulations. For simplicity, we present the combined set of days for both the WRF-GFS and WRF-NAM simulations.

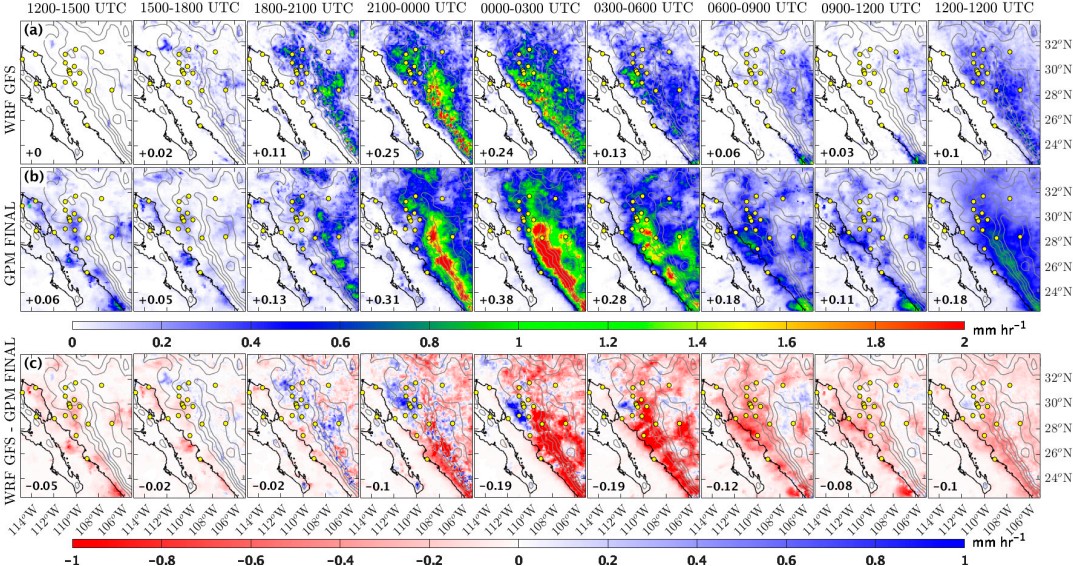

**Figure 5.** Mean hourly precipitation rate of WRF-GFS simulations (**a**), GPM Final precipitation products (**b**), and model bias (**c**) defined as WRF-GFS minus GPM Final, for combined strongly and weakly forced days within 3-hourly intervals (column 1–8) and 24-hourly intervals (column 9), are displayed. Yellow circles are observation sites of GPS Hydromet 2017. Mean values of precipitation rate across grid points (**a** and **b**) and biases (**c**) are given in the left corner of each panel. Red (blue) indicates dry (wet) model bias.

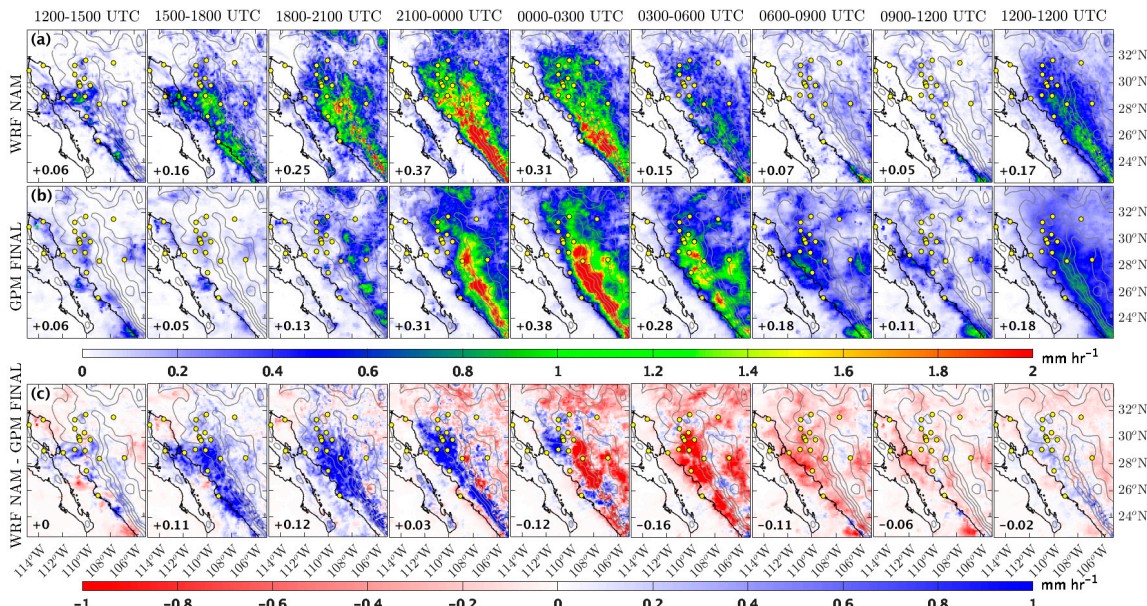

**Figure 6.** As in Figure 5, but for WRF-NAM simulations. Note that the WRF-NAM simulations exhibit wet bias in the early forecast hour compared to the WRF-GFS simulations. Mean hourly precipitation rate (**a**), GPM Final precipitation products (**b**), and model bias (**c**).

The WRF-GFS simulation (Figure 5) shows dry bias ($-0.05$ mm h$^{-1}$) in the initial period (1200–1500 UTC). This negative bias remains for the next two periods (1500–1800 UTC and 1800–2100 UTC), with mean bias values of $-0.02$ mm h$^{-1}$. The precipitation in the WRF-GFS simulation begins in the 1800–2100 UTC period in the eastern slopes and high terrain of SMO. The timing of the precipitation onset more or less matches the GPM Final precipitation estimate, as seen on the second row. However, Figure 3 shows that the model initiates precipitation about an hour late with respect to the rain gauge observations.

Starting in the 2100–0000 UTC period, the mean bias becomes more negative, especially around the mouth of GoC and the slopes of SMO to the east of TNTB. The convective systems that propagate toward the gulf and affect the coastal area (e.g., KINO and TNTB) and the western slopes of SMO are not resolved by the WRF-GFS simulation, as seen in the 0000–0300 UTC period. Instead of developing and propagating westward, the model reduces the intensity of the precipitation, as marked by a large area of negative bias occurring along the coast of the GoC, expanding from ITS1 to the south of TNTB and along the western slopes of SMO. After this period, the WRF-GFS model generates a precipitation rate less than 1 mm h$^{-1}$, as seen in 0300–0600 UTC panel. This reflects a relative lack of westward propagation of the convection. In the 0600–0900 UTC period, the WRF-GFS simulations completely terminates the precipitation in the western slopes of SMO, while the GPM Final precipitation rate estimate still shows the ongoing convective systems moving across GoC toward Baja California.

The WRF-NAM simulations (Figure 6) have high positive biases in the area around TNHM, MZTN, USMX, RAYN, and OPDE at the initial hour, which corresponds to the high WRF-NAM *PWV*, and precipitation values relative to the GPS-derived *PWV* and rain gauge measurement mentioned in the previous section. The overall mean bias across the grids in the WRF-NAM simulations is 0.11 mm hr$^{-1}$ in this early period of 1500–1800 UTC, which is much higher than that of the WRF-GFS in the same period.

When compared with the GPM Final precipitation rate estimate, the peak of the mean precipitation rate in the WRF-NAM occurs in the 2100–0000 UTC period, 3 h earlier than the GPM Final precipitation estimate. In the 0000–0300 UTC period, the WRF-NAM simulation also does not resolve convective organization and propagation westward. Instead, it reduces the precipitation rate, resulting in a dry bias, especially in the western slopes of SMO (e.g., YESX). The mean dry bias continues to increase by

4 mm h$^{-1}$ in the 0300–0600 UTC period along the eastern seaboard of GoC and the western slopes of SMO. The model does not resolve the remnant convective systems that propagate toward the GoC in the late evening to early morning hours, as captured by GPM Final precipitation estimate in the 0600–0900 UTC period.

The GPM Final precipitation estimate contains bias with respect to the rain gauge measurements, as shown in Section 2 and in sub Section 3.1. Thus, the bias calculation of the model precipitation rate, with respect to the GPM Final precipitation rate estimate, contains substantial uncertainty. We found that the WRF-GFS simulations initiate convective precipitation about an hour late with respect to rain gauge observation, but in the same hour with respect to the GPM Final precipitation estimate. Contrastingly, the WRF-NAM simulations initiate convective precipitation much earlier than rain gauge and GPM Final observations. Both WRF-GFS and WRF-NAM terminate the convective rain over the slopes of SMO too early when compared with the rain gauge and GPM Final observations. This early initiation and termination of the diurnal precipitation cycle in WRF has been found in a number of studies, such as Vincent and Lane [65] who show that the diurnal cycle of precipitation over the Maritime Continent during the Madden–Julian Oscillation passage occurs 4 to 5 h earlier than the observed precipitation cycle. The models also underestimate the westward propagation of the convective system, irrespective of the lateral boundary forcing, as also found by Moker et al. [8]. They also become drier in the period of peak precipitation (0000–0300 UTC), and at the end of the forecast hours, than the GPM Final and rain gauge measurements. A similar propagation issue has also been found by Hassim et al. [66], over New Guinea. The study shows a possible link between the propagation of the convective system and the early initiation of precipitation due to atmospheric conditions such as gravity waves, mid-level tropospheric moisture, and low-level moisture convergence.

### 4.3. Model Precipitation Skill Analysis

In Figures 7 and 8, the CSI, POD, and FAR metric differences of the WRF-GFS and the WRF-NAM models, respectively, are displayed. Each shows the precipitation forecast skills of the WRF models. Blue (red) indicates the increased forecast skills for strongly (weakly) forced days. Field significance is displayed on the bottom left and the pattern correlation between the 6-hourly and the daily forecast metrics is on the bottom right of each panel.

Both models generally have similar patterns in each time period, but there are different details in each. In the 24-h period, the models exhibit increased forecast skill in strongly forced days in the western slopes and high terrain of SMO. However, the *CSI* and *POD* differences in the WRF-GFS appear to be lesser than those in the WRF-NAM simulation, as seen in Figure 7j,k, and Figure 8j,k. The *FAR* difference between the two models is similar. This implies that the forecast skill along the western slopes of SMO during strongly forced days is better than during weakly forced days. Both models do not perform well in forecasting precipitation during weakly forced days over the western slopes of SMO and the coastal areas.

In the 1800–0000 UTC period, the *CSI* and *POD* differences in both models are statistically significant, but not the *FAR,* difference. The correlations in the *CSI* and *POD* differences between this period and the daily period are also higher than that of the *FAR*. As displayed in the *CSI*, *POD*, and FAR differences (Figure 7a–c, Figure 8a–c), the forecast skill of the WRF-GFS and WRF-NAM during strongly forced days is better than weakly forced days, particularly in the western slopes of SMO (e.g., MZTN, YESX, ITS1).

In the 0000–0600 UTC, the *CSI* and *FAR* differences of the WRF-GFS (Figure 7d,f) are not field statistically significant, while the *CSI*, *POD*, and *FAR* differences in the WRF-NAM (Figure 8d–f) are field statistically significant. The correlation values for all the differences are greater than +0.2. In general, both models exhibit better forecast skill during strongly forced days over the western slopes of SMO, and some parts of the coastal area. The *FAR* difference shows similarity for both simulations, but the WRF-NAM has higher correlation with the daily pattern.

In the 0600–1200 UTC period, the *POD* metric difference in the WRF-GFS is field statistically significant (> 90%). The other metric difference is between 80% and 90%, except the *FAR* difference in the WRF-NAM (Figure 8i). The differences of the metrics between the strongly and weakly forced days are near zero for both model simulations. High forecast skills during strongly forced days are shown in the WRF-GFS and WRF-NAM simulations over the SMO eastern slopes and its high terrain.

In summary, the WRF-GFS model exhibits higher forecast skills over the western slopes of SMO and the coastal area during strongly forced days in which the IV is present near or within the monsoon core region. The WRF-NAM model has high forecast skills over both the high terrain of SMO and the coastal areas during strongly forced days. Yet, in general, both models are more challenged to forecast and propagate MCS development during weakly forced days. This finding confirms the results of Transect 2013 in Moker et al. [8].

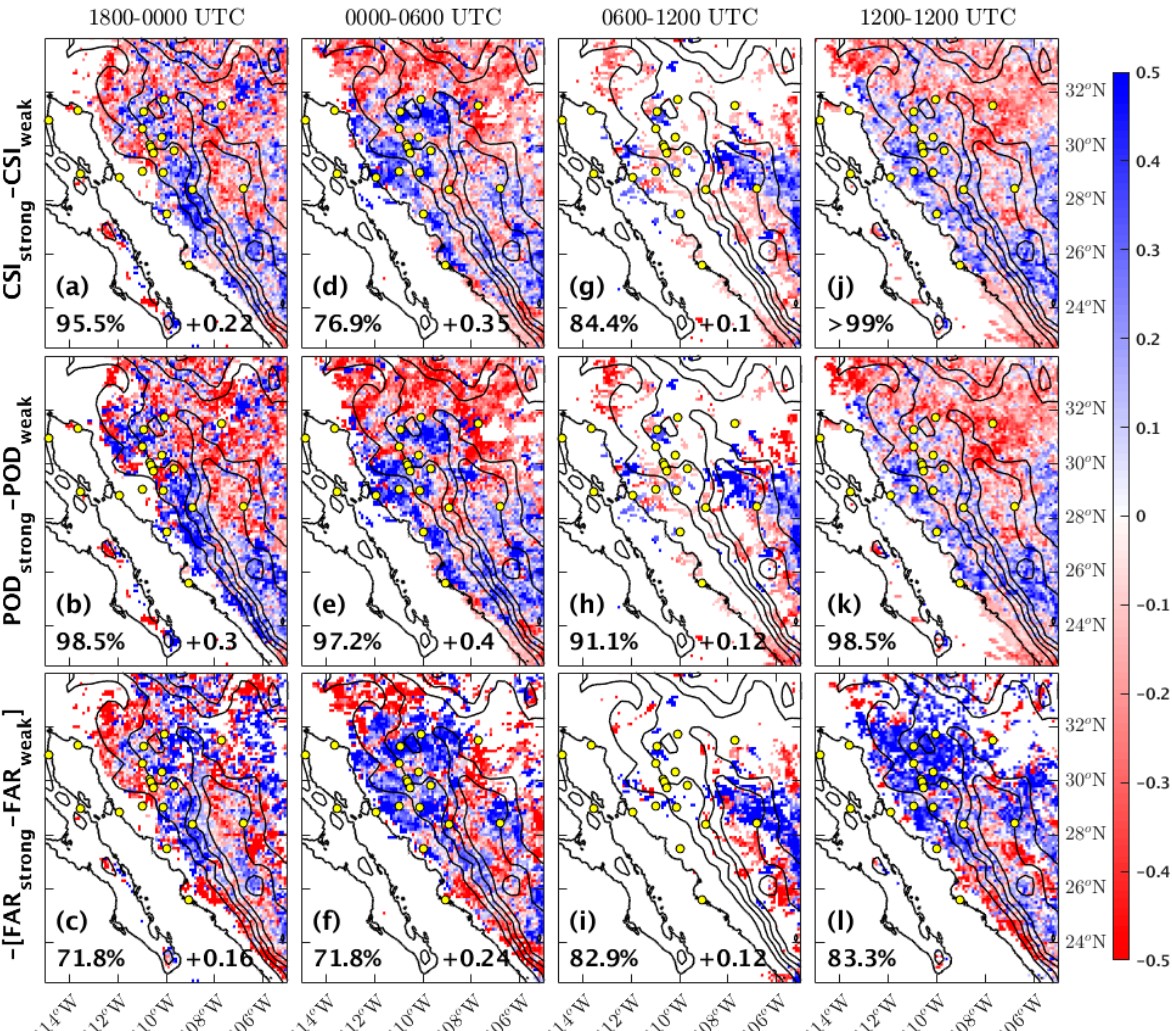

**Figure 7.** These are the *CSI* (**a,d,g,j**), *POD* (**b,e,h,k**), and *FAR* (**c,f,i,l**) metric differences between strongly and weakly forced days for the WRF-GFS simulations for the 1800–0000 UTC (**a–c**), 0000–0600 UTC (**d–f**), 0600–1200 UTC (**g–i**) and 1200–1200 UTC (**j–l**). Each metric involves a comparison between each grid point value in GPM Final and the WRF-GFS simulations. Blue (red) indicates the increased forecast skills for strongly (weakly) forced days. The statistical field significance score, displayed in the bottom left of each map, is computed using 1000 permutations in Monte Carlo resampling technique. The pattern correlation value between the 6-hourly (column a-i) and daily (j, k, l) is shown in the bottom right of each 6-hourly panel.

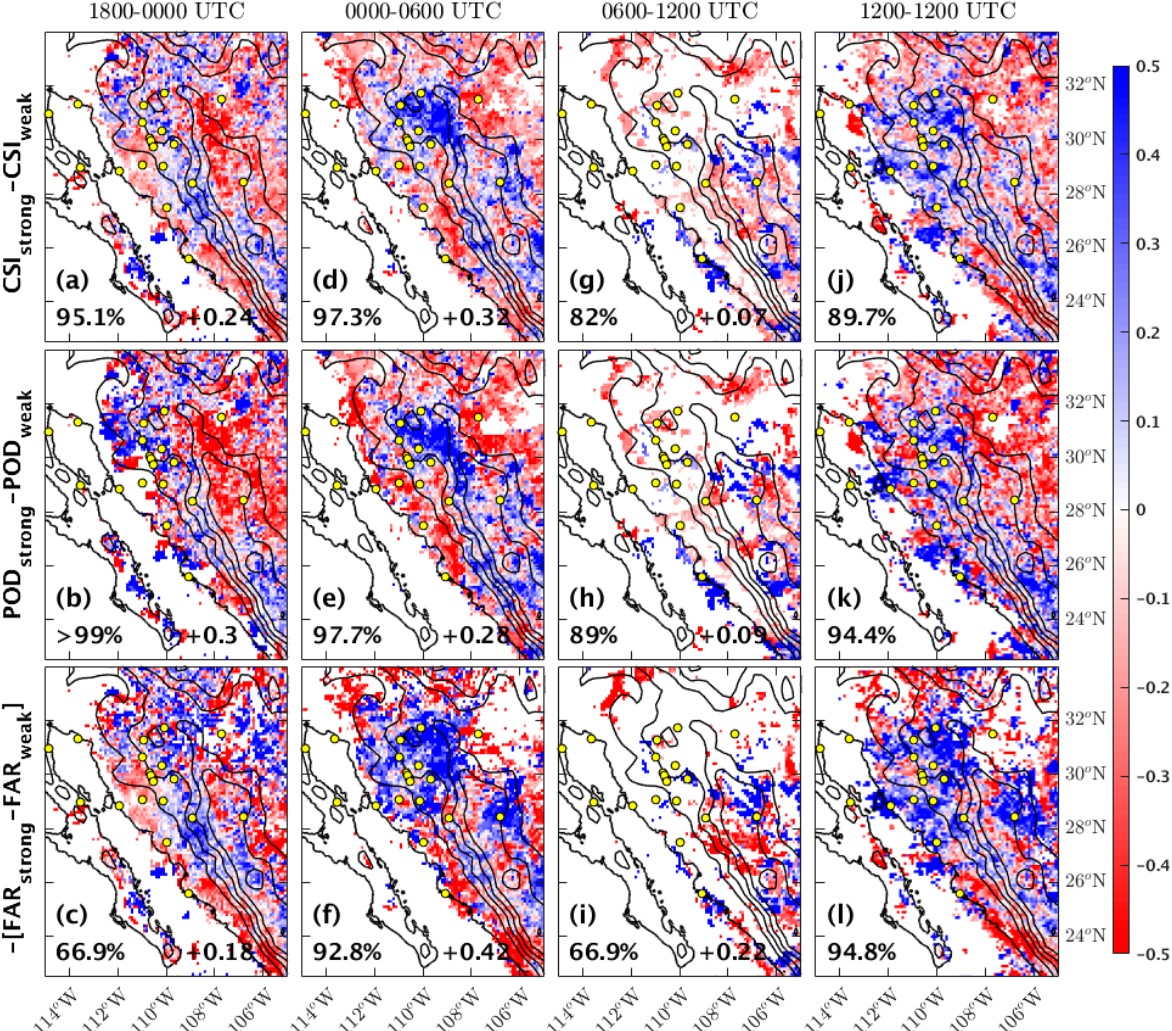

**Figure 8.** These are the *CSI* (**a,d,g,j**), *POD* (**b,e,h,k**), and *FAR* (**c,f,i,l**) metric differences between strongly and weakly forced days for the WRF-NAM simulations for the 1800–0000 UTC (**a–c**), 0000–0600 UTC (**d–f**), 0600–1200 UTC (**g–i**) and 1200–1200 UTC (**j–l**).

### 4.4. Sensitivity Analysis

Finally, we compare the mean sensitivity of the WRF-GFS *PWV*, precipitation, and QCONV across the domain, relative to the GPS-equivalent WRF-GFS *PWV* at the initial condition on the weakly forced day of 9 August, with those on the strongly forced day of 27 July. These two cases are taken for this analysis because both represent the characteristics of heavy monsoon precipitation based on the rain-gauge measurement in several sites, the extent of the convective clouds, and the dynamic pattern of the winds.

On the strongly forced day (27 July), the winds in the 2-PVU layer (dynamic tropopause) turn counterclockwise inside the monsoon core region at 1200 UTC. It is a clear indication of the IV presence. This cyclonic turning generates a PV anomaly, indicated by 2-PVU layer lowered to around 260 hPa over the northern and eastern SMO (Figure 9c). As a result, the static stability decreases due to the upward tilting of the isentropes, thus, the environment is favorable for rising motion. Mature mesoscale convective system is developed to the northeast of the IV location at 0300 UTC on 28 July, as captured by GOES 15 WV (Figure 9a). The rain gauges in TNHM and MZTN measure 21.73 mm and 10.85 mm, respectively, at 0300 UTC, and in SA80 measures 32.26 mm at 0400 UTC.

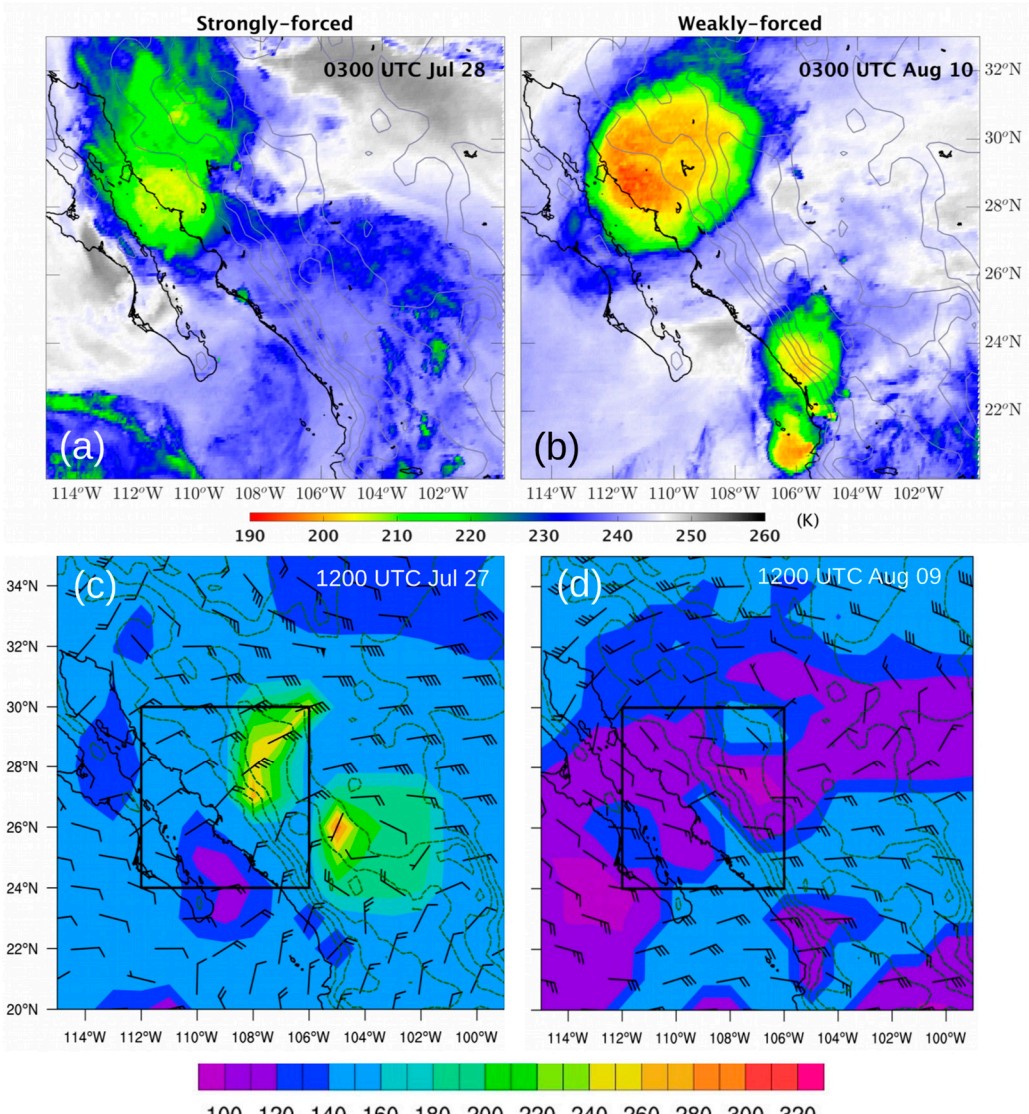

**Figure 9.** Images from GOES 15 WV are displayed on panels (**a**) and (**b**). The extent of the cloud top exceeds 100 km, and regions with temperatures less than 210 K indicate the highest points of the cloud top in the atmosphere. The pressure at 2-PVU (dynamic tropopause) and winds at 300 hPa layers taken from GFS Analysis valid at 1200 UTC are displayed on panels (**c**) and (**d**). The box indicates the monsoon core region set up by NAME 2004. Note that on the strongly forced day, the IV is characterized by the cyclonic turning of the winds and the PV anomaly at 240 to 260 hPa region inside the box.

On the weakly forced day (9 August), the 300-hPa layer does not contain cyclonic wind and the 2-PVU layer does not indicate any PV anomaly at 1200 UTC (Figure 9d). There is no IV presence. The pressure on the dynamic tropopause within the monsoon core region is almost uniform, ranging from 100 to 160 hPa. In the presence of daily surface heating and water vapor content in the atmosphere, a convective system starts to organize at 2100 UTC over the western slopes of SMO. The convective system reaches a mature stage at 0300 UTC on 10 August, with less than 190-K cloud-top temperature, indicating strong updraft occurring in the system (Figure 9b). The rain gauges at MZTN and TNHM measure 44.41 mm and 32.41 mm, respectively, from 0300 UTC to 0600 UTC. Even though the synoptic forcing is clearly different between these two days, both have clear MCSs that are of nearly identical size and occur in basically same location over Sonora.

We compute the mean sensitivity based on the topographical classification, i.e., mountainous sites and coastal sites, as shown in Figure 1 and explained in Section 3.3. The results show that the *PWV*

in the domain is more sensitive to the change in initial *PWV* across the coastal sites than across the mountainous sites, both during the weakly forced day and strongly forced day, as shown in Figure 10.

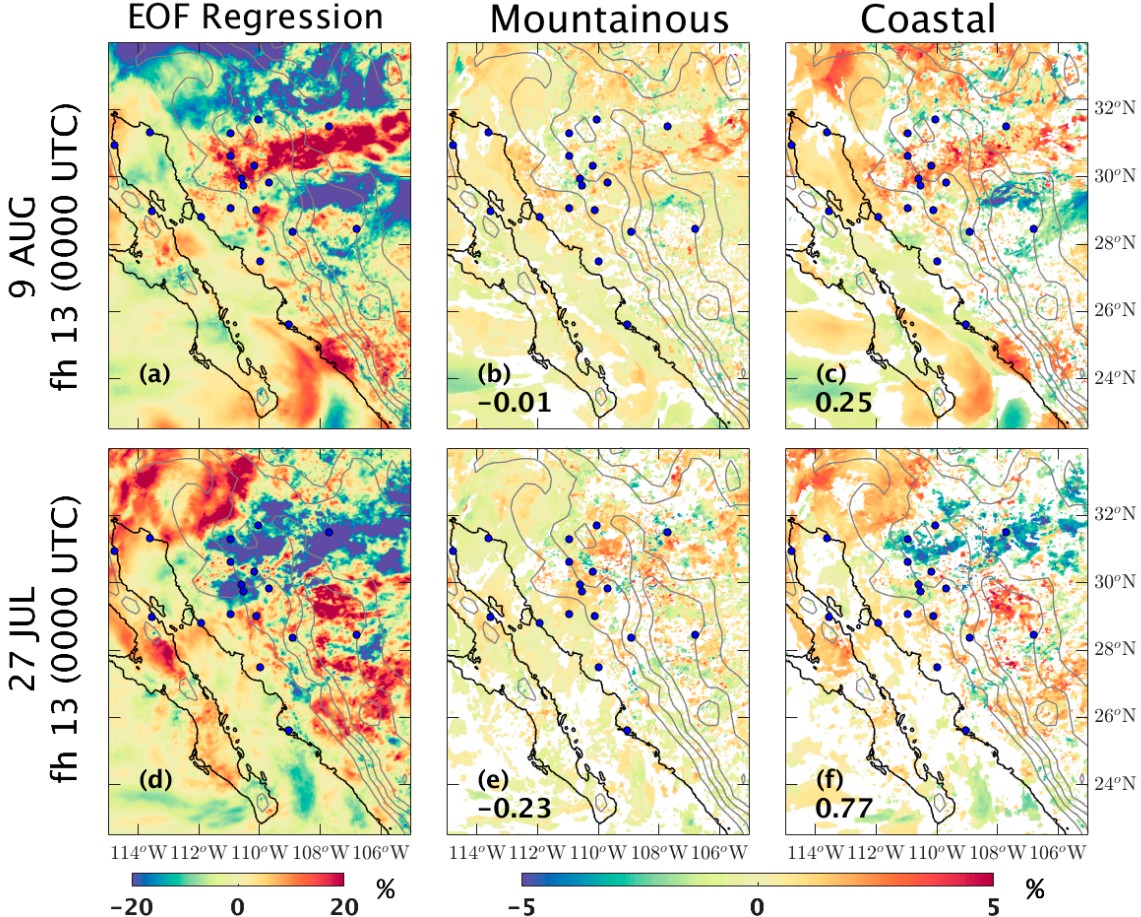

**Figure 10.** This figure displays the mean hourly sensitivity of WRF-GFS *PWV* across the domain, relative to the WRF-GFS-equivalent initial *PWV* in the mountain sites (**b**,**e**) and coastal sites (**c**,**f**) at forecast hour (fh) 13 for the 9 August (**b**,**c**) and 27 July (**e**,**f**) cases. Only statistically significant correlations ($p < 0.05$) are displayed. Regression maps of the EOF analysis are displayed for the fh 13 (**a**,**d**). Pattern correlation value between the mean hourly sensitivity and its respective EOF regression map is displayed on the lower left of the panels (**b**,**c**,**e**,**f**). Note that the *PWV* in the domain is more sensitive to the change in initial *PWV* in the coastal sites than that in the mountainous site. The pattern of the coastal areas resembles the regression map for each case, as shown by the pattern correlation values.

On the weakly forced day, the *PWV* is most sensitive over the northern part of SMO (Figure 10c). A hundred mm change of initial *PWV* in the coastal sites produces a 5 mm or more increase of *PWV* in the area at those hours. A highly negative sensitivity occurs at 0000 UTC in the eastern slopes of SMO. This means that a 100 mm change in initial *PWV* in the coastal sites generates around a 5 mm or more decrease of *PWV* in the area at that hour. On the strongly forced day, the *PWV* is negatively sensitive across the northern part of SMO. This means that a 100 mm change of initial *PWV* in the coastal sites reduces the *PWV* in the region by around 5 mm (Figure 10f).

The statistically significant dominant mode, generated by the covariance-based EOF analysis (Figure 10a,d) at 0000 UTC, agrees with the mean sensitivity of *PWV* for both weakly and strongly forced days, in Figure 10c,f, respectively. The pattern in Figure 10a resembles the pattern in Figure 10c, with a correlation value of 0.25. The area of positive percentage is situated across the northern SMO, stretching to east-northeast with area of negative percentage just to the south. The covariance-based

EOF analysis confirms that the WRF-GFS-equivalent initial *PWV* in the coastal sites influences the *PWV* across the domain in the later forecast hours more than the initial *PWV* in the mountainous sites.

For the strongly forced day, the pattern in Figure 10d resembles the pattern in Figure 10f, with a pattern correlation value of 0.77. The area with negative percentage is spreading west to east over the northern SMO, with positive percentage area to the north and the south. Similar to the weakly forced day, the covariance-based EOF analysis for the 27 July case shows that the WRF-GFS-equivalent initial *PWV* in the coastal sites affects the *PWV* in the domain in the later forecast hours more than the initial *PWV* in the mountainous sites. Therefore, the dominant mode of sensitivity separates the mean sensitivity of the mountainous sites from the coastal sites.

The sensitivity analyses for the precipitation produces contrary results when compared to the *PWV*. The change of precipitation across the domain in the later forecast hours during the weakly forced day is more sensitive to the change of the initial *PWV* in the mountainous sites than that in the coastal sites, for both strongly and weakly forced days (Figure 11).

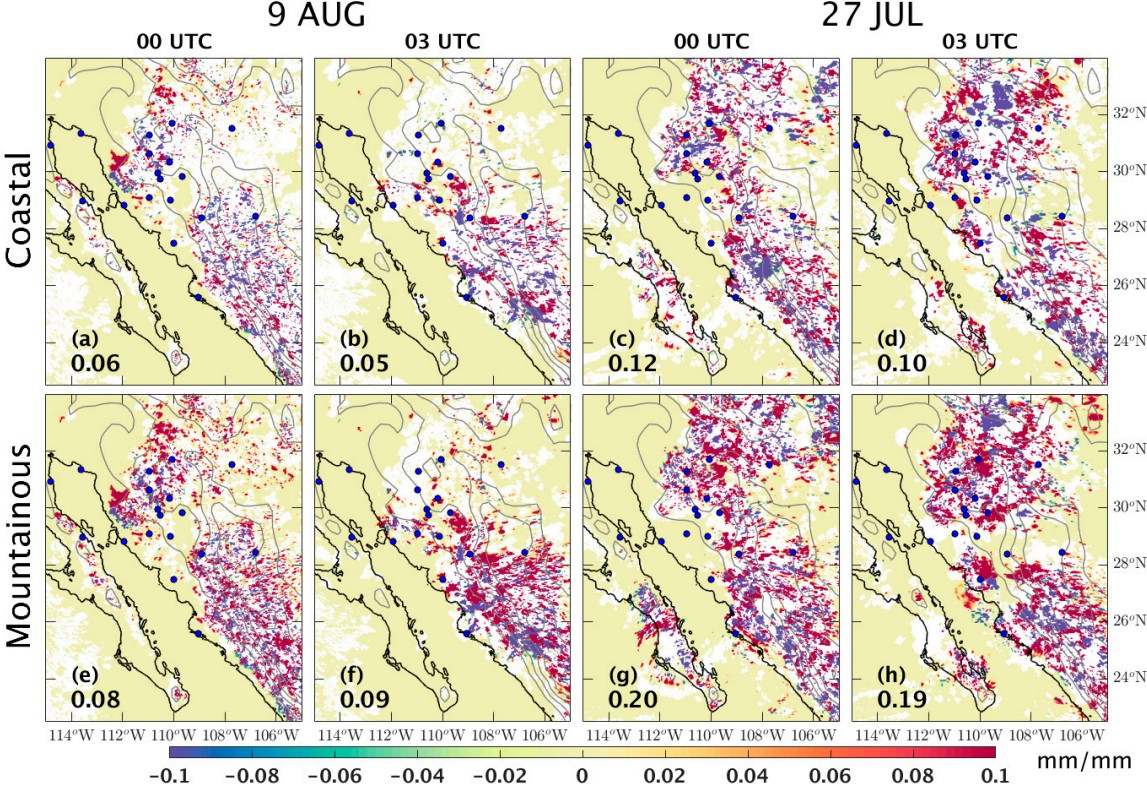

**Figure 11.** This figure shows the mean hourly sensitivity of WRF-GFS precipitation across the domain, relative to the WRF-GFS-equivalent initial *PWV* in the coastal and mountainous sites, at 0000 UTC and 0300 UTC for both weakly (**a**, **b**, **e**, **f**) and strongly (**c**, **d**, **g**, **h**) forced days. Only statistically significant correlations (p < 0.05) are displayed. The mean of the absolute values across the domain is shown on the lower left of each panel. The mountainous sites exhibit higher values than the coastal sites on both weakly and strongly forced days.

The impact of the initial *PWV* change in the mountainous sites versus the coastal sites to the precipitation across the domain during the weakly forced day is shown around the peak of precipitation, at 0000 UTC and 0300 UTC (Figure 11a,b,e,f). Over the western slopes of SMO, a millimeter change of initial *PWV* in the mountainous sites generates a 0.1 mm or more increase of precipitation around USMX, MZTN, ITS1, YESX, TNCU, and over SMO in general (Figure 11e,f). The change in initial *PWV* in the coastal sites (Figure 11a,b), on the other hand, does not generate as great an increase as the mountainous sites. Similarly, on the strongly forced day, the change in precipitation across the

domain is more sensitive to the change in initial *PWV* in the mountainous sites (Figure 11g,h) than in the coastal sites (Figure 11c,d). The sensitivity difference is clear over the northern slopes of SMO.

The mean of absolute values across the domain on each panel quantitatively indicates the distinct influence of the model-equivalent initial *PWV* of each class on the change of precipitation across the domain. The mean sensitivity for the mountainous sites is higher than that for the coastal sites. This is true for both weakly and strongly forced days. These quantitative values confirm that the WRF-GFS-equivalent initial *PWV* in the mountainous sites induces a greater influence on the change of precipitation across the domain in the later forecast hours than it does in the coastal sites. In a larger context, this fact also means that the convective organization and propagation across the domain is more sensitive to the initial *PWV* over the mountainous areas than the coastal areas.

The sensitivity analyses of QCONV (Figure 12) are consistent with those of precipitation. The QCONV across the domain is sensitive to the change in initial *PWV* in the mountainous sites more than in the coastal sites during the weakly and strongly forced days. The difference is particularly visible from 0000 UTC and 0300 UTC, where the increase of QCONV is situated around KINO, TNHM, MZTN, YESX, and ITS1, on the western slopes of SMO.

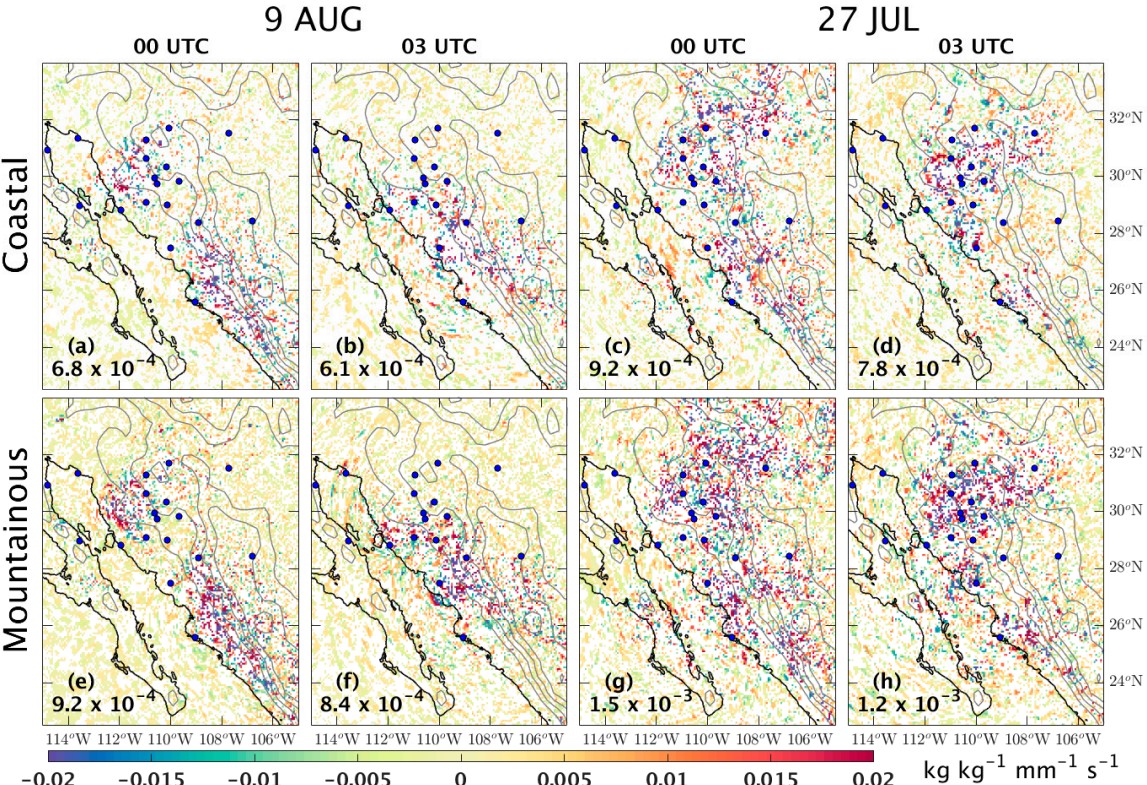

**Figure 12.** This figure displays the mean hourly sensitivity of WRF-GFS QCONV across the domain, relative to the WRF-GFS-equivalent initial *PWV* in the coastal and mountainous sites, at 0000 UTC and 0300 UTC for both weakly (**a,b,e,f**) and strongly (**c,d,g,h**) forced days. Smoothing is applied by scaling up the grid spacing from 2.5 km to 1/16°. Only statistically significant correlations (p < 0.05) are displayed. The mean of absolute value across the domain is displayed on the lower left on each panel. The mountainous sites exhibit higher values than the coastal sites on both weakly and strongly forced days.

On the weakly forced day, a millimeter change of initial *PWV* in the mountainous sites creates an increase of QCONV as much as 0.02 kg kg$^{-1}$ s$^{-1}$ (Figure 12e,f). In the mean QCONV sensitivity across the coastal sites (Figure 12a,b), the increase signature of QCONV is hardly apparent. However, the mean of the absolute values across the domain reveals that the change of WRF-GFS-equivalent initial *PWV* over the mountainous sites affects the change of QCONV across the domain more it does over

the coastal sites. During the peak of the monsoon precipitation around 0000 UTC, the mean of absolute values of the QCONV sensitivity, relative to the WRF-GFS-equivalent initial *PWV* over mountainous sites, is $9.2 \times 10^{-4}$ kg kg$^{-1}$ mm$^{-1}$ s$^{-1}$, whereas the mean of absolute values of sensitivity relative to the WRF-GFS-equivalent initial *PWV* over the coastal site is only $6.8 \times 10^{-4}$ kg kg$^{-1}$ mm$^{-1}$ s$^{-1}$.

As seen from this pattern, there is minimal difference in QCONV sensitivity between the mountainous (Figure 12g,h) and coastal sites (Figure 12c,d) during the strongly forced day. The mean of absolute values across the domain at 0000 UTC, for example, is $1.5 \times 10^{-3}$ kg kg$^{-1}$ mm$^{-1}$ s$^{-1}$ for the sensitivity, relative to the change in WRF-GFS-equivalent initial *PWV* in the mountainous sites, and $9.2 \times 10^{-4}$ kg kg$^{-1}$ mm$^{-1}$ s$^{-1}$ for the sensitivity relative to the change in WRF-GFS-equivalent initial *PWV* in the coastal sites. Similar to the weakly forced day, the change in QCONV across the domain during the strongly forced day is more sensitive to the change in WRF-GFS-equivalent initial *PWV* in the mountainous sites than in the coastal sites. Therefore, mountainous regions matter in terms of convective initiation and precipitation.

There are two questions to answer: (1) Why does changing initial *PWV* in the coastal sites generate changes in *PWV* across the domain more than changes in the mountainous sites? (2) Why does changing initial *PWV* in the mountainous sites produce more changes in precipitation and QCONV across the domain than changes in the coastal sites? To answer the first question, we hypothesize that there is positive correlation between the initial *PWV* in the coastal sites and *PWV* across the domain, since the GoC is the moisture source of the domain. Therefore, the WRF-GFS model confirms what has been found by Adams and Comrie [4] that the moisture during NAMS partly comes from the GoC, and, thus, an increase in *PWV* in the coastal site will result in an increase of *PWV* across the domain in the later forecast hours.

For the second question, the answer is that mountainous topography naturally induces convective initiation. Daily solar heating on its surface develops thermal differences between the mountain and the surrounding atmosphere, and upslope winds during daytime help water vapor to rise and condense. Both mechanisms create instability and assist convective initiation over mountainous regions. Thus, an increase in *PWV* in the mountainous sites does increase the precipitation and QCONV on the western slopes of SMO and the northern part of SMO, as shown in the Figures 11 and 12. Moreover, since the convective system propagates westward, the change in initial *PWV* in the mountainous sites also affects the change in precipitation and QCONV in the coastal areas at the later forecast hours. Similar results appear in other modeling studies, showing that an increase in precipitation in mountainous regions, such as Kilimanjaro and the Andes mountain ranges in Chile, is closely related to a change in the ambient condition and variables such as the geometry of the location, with respect to wind flows, as well as an increase in water vapor and temperature [60,67–69].

## 5. Conclusions

The WRF-ARW simulations for daily monsoon precipitation events during the GPS Hydromet 2017 field campaign are performed with lateral boundary forcing and initial conditions from the GFS and NAM models. We classify the monsoon precipitation events into two: (1) strongly forced days in which an IV is present near or within the monsoon core region, and (2) weakly forced days when IV is absent. GPM Final precipitation product is used as the ground reference.

Compared to the GPS-derived *PWV*, the WRF-GFS and WRF-NAM *PWV* exhibits wet biases at the initial forecast hour. While WRF-GFS *PWV* is still within the standard error of the observation, WRF-NAM *PWV* exceeds the standard error most of the time. The diurnal cycle of both WRF-GFS and WRF-NAM is out of phase with time, compared to that of the GPS-derived *PWV*. The phase problem is due to the rapid decrease and increase after 6 h into the forecast. The model *PWV* diurnal cycles also correspond to the model precipitation diurnal cycles. The high WRF-NAM *PWV* values in the early forecast hours are associated with the early precipitation in WRF-NAM, and the rapid decrease in model *PWV* is related to the early termination of model precipitation in both WRF-GFS and WRF-NAM.

Both model simulations have timing issues and discrepancy in accumulated precipitation diurnal cycles, as compared to the observed precipitation in the GPM Final precipitation estimate and rain gauge measurement. The WRF-NAM simulations initiate precipitation as early as the initial forecast hour. The *PWV* positive biases at the initial condition are likely the cause of the early precipitation in the WRF-NAM simulations. The WRF-GFS precipitation diurnal cycle is closer to that of the rain gauge than to that of the GPM Final precipitation estimate. Yet, both models terminate precipitation too early (in the 0300–0600 UTC period), while the GPM Final precipitation and rain gauge measurements still show ongoing precipitation. Thus, model precipitation exhibits dry bias starting in the 0000–0300 UTC period, regardless of its boundary forcing. Lastly, both WRF-GFS and WRF-NAM underestimate the westward propagation of MCSs at the end of the daily convective cycle. However, we need to take into account the uncertainty of the GPM Final precipitation estimate, since it underestimates the precipitation relative to the rain gauge measurement.

Model accuracy in forecasting precipitations is evaluated using *CSI*, *POD*, and *FAR* metrics. The difference of the *CSI*, *POD*, and *FAR* scores between the strongly forced day and weakly forced day simulations are computed. Both models exhibit higher precipitation forecast skill during the strongly forced days, especially over the high terrain and the western slopes of SMO, as well as the coastal area on the eastern seaboard of the GoC. Yet both models are more challenged to forecast MCS development and propagation during weakly forced days.

The WRF-GFS *PWV* across the domain during the weakly forced day of 9 August is more sensitive to the change in initial *PWV* in the coastal sites than that in the mountainous sites. During the strongly forced day of 27 July, even though the difference in sensitivity between the coastal and mountainous sites is minimal, the WRF-GFS *PWV* across the domain is more sensitive to the change in initial *PWV* in the coastal sites. In contrast, the WRF-GFS precipitation and QCONV across the domain during weakly and strongly forced days are more sensitive to the change in initial *PWV* in the mountainous sites than the coastal sites. The reason for precipitation more sensitive to the *PWV* in the mountain areas is that the mountain topography helps the thermodynamics and the dynamics of convective initiation.

Data assimilation will be the next step of this study. Besides improving the initialization of the physical state variables, such as *PWV*, putting more weight on the GPS sites in the mountainous areas in the data assimilation may improve the monsoon precipitation forecast in the region during weakly forced days. Future research may be directed toward resolving the time lags in the WRF *PWV* diurnal cycle and the WRF precipitation initiation and termination.

**Supplementary Materials:** The following are available online at http://www.mdpi.com/2073-4433/10/11/694/s1, Figure S1: Mean diurnal cycle of WRF-GFS *PWV* and GPS-derived *PWV* (or observations) at MGDA site for combined strongly–weakly forced days (a), weakly forced days (b), and strongly forced days (c). Mean diurnal cycle of precipitation at MGDA site based on WRF-GFS simulations, rain gauge measurement, and GPM Final product for combined strongly–weakly forced days (d), weakly forced days (e), and strongly forced days (f). Note the *PWV* maximum occurs almost at the same time as the peak of precipitation, Figure S2: Mean diurnal cycle of WRF-GFS *PWV* and GPS-derived *PWV* (or observations) at SA80 site for combined strongly–weakly forced days (a), weakly forced days (b), and strongly forced days (c). Mean diurnal cycle of precipitation at MGDA site based on WRF-GFS simulations, rain gauge measurement, and GPM Final product for combined strongly–weakly forced days (d), weakly forced days (e), and strongly forced days (f). Note the *PWV* maximum occurs almost at the same time as the peak of precipitation.

**Author Contributions:** C.B.R. developed all the figures and wrote the paper. C.B.R., J.M.M.J., and L.M.F. had responsibility of data analysis, model configuration, and simulations. C.L.C. and A.F.A.J. conceived of the project and research, and reviewed and edited the paper. D.K.A. provided the research with ground-based data. C.M.S. organized the data collection.

**Funding:** GPS Hydromet 2017 field campaign and post-field campaign research were funded by Binational Consortium for Regional Scientific Development and Innovation at the University of Arizona and the Consejo Nacional de Ciencia y Technología de México.

**Acknowledgments:** We thank Camacho Pérez Santiago of CONAGUA and Enrico Yepez of Instituto Technológico de Sonora for providing additional precipitation dataset.

**Conflicts of Interest:** The authors declare no conflict of interest.

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
