# Peer review of "Evaluating Forecast Skills of Moisture from Convective-Permitting WRF-ARW Model during 2017 North American Monsoon Season"

_atmosphere, doi:10.3390/atmos10110694_

Round 1
Reviewer 1 Report
Review of ‘Evaluating Forecast Skills of Moisture from Convective-Permitting WRF-ARW Model during 2017 North American Monsoon Season’ by Risanto et al.
General:
This is a very thorough and well written paper, which evaluates the simulated precipitable water, precipitation and convective organisation (through propagating squall lines). Although the study is very thorough, I have a few questions about the role of cloud moisture in the simulations - the authors have examined water vapour and surface precipitation, but what is going on with the simulated cloud hydrometeors (which may or may not land on the surface as rain)? Also, I would like to understand better whether the double peaks in rainfall and PWV are due to an actual double peak in rainfall on some days, or due to some days having an early peak and some days having a later peak? If it is a genuine double peak on some days, what is the role of stratiform rain later in the day?
Specific comments
1. Is the WRF early initiation only due to the model initialisation? A number of studies (eg Vincent and Lane 2016) have shown that the diurnal cycle in WRF tends to peak too early in the day, even in long runs that are not reinitialised. A related question is what the role of forecast spin-up is in your simulations. What happens if they are initialised 6 hours earlier?
2. I find the mention of data assimilation in the last sentence of the abstract a little misplaced, since you didn’t to this in this paper.
3. There is an obvious challenge in knowing the ‘truth’ in all three data sets. The WRF values are some kind of spatial average over the grid box, as are the satellite values, so neither are really representative of each other or the stations. Since you have quite a lot of stations, would it be possible to average the station data over a well-resolved area to compare with an analogous average from the model and satellite?
4. I believe the definition of specific humidity is not strictly correct - it should be mv / (mv + md), while the mixing ratio given by WRF is mv / md - so your specific humidity should be Qv / (1+Qv).
5. Line 320 - which variables were perturbed, and how?
6. Line 321 - how did you decide what was ‘appropriate variance’?
7. Line 353 - I’m not convinced by your definition of ‘coastal’ as everything below 500m. I would have expected three regions - mountains, coastal and low-lying land.
8. Maybe there is a link between the early initiation of precipitation, and the errors in westward propagating precipitation, if latent heating plays a role in initiating diurnally forced gravity waves (see Hassim et al. (2016)
9. Line 300 - replace ‘follow’ with ‘follows’
10. Line 369 - something wrong with this sentence. Replace with ‘The simulation is initialised at 1200 UTC (0500 LT) and runs for 24 hours’.
Vincent, C.L. and T.P. Lane, 2016: Evolution of the Diurnal Precipitation Cycle with the Passage of a Madden–Julian Oscillation Event through the Maritime Continent. Mon. Wea. Rev., 144, 1983–2005
Hassim, M. E. E., Lane, T. P., and Grabowski, W. W.: The diurnal cycle of rainfall over New Guinea in convection-permitting WRF simulations, Atmos. Chem. Phys., 16, 161–175, https://doi.org/10.5194/acp-16-161-2016, 2016.
Reviewer 2 Report
The authors present convection-permitting ensemble simulations of the North American Monsoon, they compare the results using satellite and station data before conducting an ensemble sensitivity analysis for the lowest resolution configuration over a 24 h period.
The manuscript is well written and provides a large amount of information.
There is a lot of material presented which has resulted in redundancy, for example figures 2,3 and table 3 practically show the same information. I feel this manuscript could be improved by loosing the redundancy and shortening some of the figures (and hence description) to the main points. As another example we do not need to see maps of the metric differences every hour as currently shown in some of the figures.
I also have a couple of minor comments
Make sure you define QCONV and PWV in the main text. Make sure that all figures have clearly defined panels (e.g. Fig. 1 does not)Author Response
Please see the attachment.
